# DoWG Unleashed: An Efficient Universal Parameter-Free Gradient Descent Method

**Ahmed Khaled**
Princeton University

**Konstantin Mishchenko**
Samsung AI Center

**Chi Jin**
Princeton University

## Abstract

This paper proposes a new easy-to-implement *parameter-free* gradient-based optimizer: DoWG (Distance over Weighted Gradients). We prove that DoWG is *efficient*—matching the convergence rate of optimally tuned gradient descent in convex optimization up to a logarithmic factor without tuning any parameters, and *universal*—automatically adapting to both smooth and nonsmooth problems. While popular algorithms following the AdaGrad framework compute a running average of the squared gradients to use for normalization, DoWG maintains a new distance-based weighted version of the running average, which is crucial to achieve the desired properties. To complement our theory, we also show empirically that DoWG trains at the edge of stability, and validate its effectiveness on practical machine learning tasks.

## 1 Introduction

We study the fundamental optimization problem

$$\min_{x \in \mathcal{X}} f(x), \tag{OPT}$$

where $f$ is a convex function, and $\mathcal{X}$ is a convex, closed, and bounded subset of $\mathbb{R}^d$. We assume $f$ has at least one minimizer $x_* \in \mathcal{X}$. We focus on gradient descent and its variants, as they are widely adopted and scale well when the model dimensionality $d$ is large (Bottou et al., 2018). The optimization problem (OPT) finds many applications: in solving linear systems, logistic regression, support vector machines, and other areas of machine learning (Boyd and Vandenberghe, 2004). Equally important, methods designed for (stochastic) convex optimization also influence the intuition for and design of methods for nonconvex optimization– for example, momentum (Polyak, 1964), AdaGrad (Duchi et al., 2010), and Adam (Kingma and Ba, 2015) were all first analyzed in the convex optimization framework.

As models become larger and more complex, the cost and environmental impact of training have rapidly grown as well (Sharir et al., 2020; Patterson et al., 2021). Therefore, it is vital that we develop more efficient and effective methods of solving machine learning optimization tasks. One of the chief challenges in applying gradient-based methods is that they often require tuning one or more stepsize parameters (Goodfellow et al., 2016), and the choice of stepsize can significantly influence a method's convergence speed as well as the quality of the obtained solutions, especially in deep learning (Wilson et al., 2017).

The cost and impact of hyperparameter tuning on the optimization process have led to significant research activity in designing parameter-free and adaptive optimization methods in recent years, see e.g. (Orabona and Cutkosky, 2020; Carmon and Hinder, 2022) and the references therein.

We say an algorithm is *universal* if it adapts to many different problem geometries or regularity conditions on the function $f$ (Nesterov, 2014; Levy et al., 2018; Grimmer, 2022). In this work, we focus on two regularity conditions: (a) $f$ Lipschitz and (b) $f$ smooth. Lipschitz functions

have a bounded rate of change, that is, there exists some $G > 0$ such that for all $x, y \in \mathcal{X}$ we have $|f(x) - f(y)| \leq G\|x - y\|$. The Lipschitz property is beneficial for the convergence of gradient-based optimization algorithms. They converge even faster on smooth functions, which have continuous derivatives; that is, there exists some $L > 0$ such that for all $x, y \in \mathcal{X}$ we have $\|\nabla f(x) - \nabla f(y)\| \leq L\|x - y\|$. Smoothness leads to faster convergence of gradient-based methods than the Lipschitz property. Universality is a highly desirable property because in practice the same optimization algorithms are often used for both smooth and nonsmooth optimization (e.g. optimizing both ReLU and smooth networks).

The main question of our work is as follows:

Can we design a universal, parameter-free gradient descent method for (OPT)?

Existing universal variants of gradient descent either rely on line search (Nesterov, 2014; Grimmer, 2022), bisection subroutines (Carmon and Hinder, 2022), or are not parameter-free (Hazan and Kakade, 2019; Levy et al., 2018; Kavis et al., 2019). Line search algorithms are theoretically strong, achieving the optimal convergence rates in both the nonsmooth and smooth settings with only an extra log factor. Through an elegant application of bisection search, Carmon and Hinder (2022) design a parameter-free method whose convergence is only double-logarithmically worse than gradient descent with known problem parameters. However, this method requires resets, i.e. restarting the optimization process many times, which can be very expensive in practice. Therefore, we seek a universal, parameter-free gradient descent method for (OPT) with **no search subroutines.**

**Our contributions.** We provide a new algorithm that meets the above requirements. Our main contribution is **a new universal, parameter-free gradient descent method with no search subroutines.** Building upon the recently proposed Distance-over-Gradients (DoG) algorithm (Ivgi et al., 2023), we develop a new method, DoWG (Algorithm 1), that uses a different stepsize with adaptively weighted gradients. We show that DoWG automatically matches the performance of gradient descent on (OPT) up to logarithmic factors with no stepsize tuning at all. This holds in both the nonsmooth setting (Theorem 3) and the smooth setting (Theorem 4). Finally, we show that DoWG is competitive on real machine learning tasks (see Section 4).

| Algorithm | No search | Parameter-free | Universal | GD framework |
|---|---|---|---|---|
| Polyak stepsize (Polyak, 1987; Hazan and Kakade, 2019) | ✓ | ✗ | ✓ | ✓ |
| Coin betting with normalization (Orabona and Pál, 2016; Orabona and Cutkosky, 2020; Orabona, 2023) | ✓ | ✓ | ✓[*] | ✗ |
| Nesterov line search (Nesterov, 2014) | ✗ | ✓ | ✓ | ✓ |
| AdaGrad (Duchi et al., 2010; Levy et al., 2018; Ene et al., 2021) | ✓ | ✗ | ✓ | ✓ |
| Adam (Kingma and Ba, 2015; Li et al., 2023) | ✓ | ✗ | ✓ | ✓ |
| Bisection search (Carmon and Hinder, 2022) | ✗ | ✓ | ✓ | ✓ |
| D-Adaptation (Defazio and Mischenko, 2023) | ✓ | ✓ | ✗ | ✓ |
| DoG (Ivgi et al., 2023) | ✓ | ✓ | ✓[*] | ✓ |
| DoWG (**new, this paper!**) | ✓ | ✓ | ✓ | ✓ |

[*] Result appeared after the initial release of this paper.
Table 1: A comparison of different adaptive algorithms for solving (OPT). "Universal" means that the algorithm can match the rate of gradient descent on both smooth and nonsmooth objectives up to polylogarithmic factors. "No search" means the algorithm does not reset. "GD framework" refers to algorithms that follow the framework of Gradient Descent.

## 2  Related Work

There is a lot of work on adaptive and parameter-free approaches for optimization. We summarize the main properties of the algorithms we compare against in Table 1. We enumerate some of the major approaches below:

**Polyak stepsize.** When $f_* = f(x_*)$ is known, the Polyak stepsize (Polyak, 1987) is a theoretically-grounded, adaptive, and universal method (Hazan and Kakade, 2019). When $f_*$ is not known, Hazan and Kakade (2019) show that an adaptive re-estimation procedure can recover the optimal convergence rate up to a log factor when $f$ is Lipschitz. Loizou et al. (2021) study the Polyak stepsize in stochastic non-convex optimization. Orvieto et al. (2022) show that a variant of the Polyak stepsize with decreasing stepsizes can recover the convergence rate of gradient descent, provided the stepsize is initialized properly. Unfortunately, this initialization requirement makes the method not parameter-free.

**The doubling trick.** The simplest way to make an algorithm parameter-free is the doubling-trick. For example, for gradient descent for $L$-smooth and convex optimization, the stepsize $\eta = \frac{1}{L}$ results in the convergence rate of

$$f(\hat{x}) - f_* = \mathcal{O}\left(\frac{D_0 L^2}{T}\right), \tag{1}$$

where $D_0 = \|x_0 - x_*\|$. We may therefore start with a small estimate $L_0$ of the smoothness constant $L$, run gradient descent for $T$ steps, and return the average point. We restart and repeat this for $N$ times, and return the point with the minimum function value. So long as $N \geq \log \frac{L}{L_0}$, we will return a point with loss satisfying eq. (1) at the cost of only an additional logarithmic factor. This trick and similar variants of it appear in the literature on prediction with expert advice and online learning (Cesa-Bianchi et al., 1997; Cesa-Bianchi and Lugosi, 2006; Hazan and Megiddo, 2007). It is not even needed to estimate $N$ in some cases, as the restarting can be done adaptively (Streeter and McMahan, 2012). In practice, however, the performance of doubling trick suffers from restarting the optimization process and throwing away useful that could be used to guide the algorithm.

**Parameter-free methods.** Throughout this paper, we use the term "parameter-free algorithms" to describe optimization algorithms that do not have any tuning parameters. We specifically consider only the deterministic setting with a compact domain. As mentioned before, Carmon and Hinder (2022) develop an elegant parameter-free and adaptive method based on bisection search. Bisection search, similar to grid search, throws away the progress of several optimization runs and restarts, which may hinder their practical performance. Ivgi et al. (2023); Defazio and Mishchenko (2023) recently developed variants of gradient descent that are parameter-free when $f$ is Lipschitz. However, D-Adaptation (Defazio and Mishchenko, 2023) has no known guarantee under smoothness, while DoG (Ivgi et al., 2023) was only recently (after the initial release of this paper) shown to adapt to smoothness. We compare against the convergence guarantees of DoG in Section 3.2. For smooth functions, Malitsky and Mishchenko (2020) develop AdGD, a method that efficiently estimates the smoothness parameter on-the-fly from the training trajectory. AdGD is parameter-free and matches the convergence of gradient descent but has no known guarantees for certain classes of Lipschitz functions. A proximal extension of this method has been proposed by Latafat et al. (2023).

**Parameter-free methods in online learning.** In the online learning literature, the term "parameter-free algorithms" was originally used to describe another class of algorithms that adapt to the unknown distance to the optimal solution (but can still have other tuning parameters such as Lipschitz constant). When the Lipschitz parameter is known, approaches from online convex optimization such as coin betting (Orabona and Pál, 2016), exponentiated gradient (Streeter and McMahan, 2012; Orabona, 2013), and others (McMahan and Orabona, 2014; Orabona and Cutkosky, 2020; Orabona and Pál, 2021; Orabona and Tommasi, 2017) yield rates that match gradient descent up to logarithmic factors. Knowledge of the Lipschitz constant can be removed either by using careful restarting schemes (Mhammedi et al., 2019; Mhammedi and Koolen, 2020), or adaptive clipping on top of coin betting (Cutkosky, 2019). For optimization in the deterministic setting, it is later clarified that, by leveraging the normalization techniques developed in (Levy, 2017), the aforementioned online learning algorithms can be used without knowing other tuning parameters (i.e., achieve "parameter-free" in the sense of this paper) for optimizing both Lipschitz functions (Orabona and Pál, 2021) and smooth functions (Orabona, 2023). Concretely, as shown in Orabona (2023) (which appears after the initial release of this paper), combining algorithms in Streeter and McMahan (2012); Orabona and Pál (2016) with normalization techniques (Levy, 2017) yields new algorithms that are also search-free, parameter-free (in the sense of this paper), and universal. However, these algorithms are rather different from DoWG in algorithmic style: these algorithms only use normalized gradients while DoWG does use the magnitudes of the gradients; DoWG falls in the category of gradient descent algorithms with adaptive learning rate, while these algorithms do not.

**Line search.** As mentioned before, line-search-based algorithms are universal and theoretically grounded (Nesterov, 2014) but are often expensive in practice (Malitsky and Mishchenko, 2020).

**AdaGrad family of methods.** Li and Orabona (2019) study a variant of the AdaGrad stepsizes in the stochastic convex and non-convex optimization and show convergence when the stepsize is tuned to depend on the smoothness constant. Levy et al. (2018) show that when the stepsize is tuned properly to the diameter of the domain $\mathcal{X}$ in the constrained convex case, AdaGrad-Norm adapts to smoothness. Ene et al. (2021) extend this to AdaGrad and other algorithms, and also to variational inequalities. Ward et al. (2019); Traoré and Pauwels (2021) show the convergence of AdaGrad-Norm for any stepsize for non-convex (resp. convex) optimization, but in the worst case the dependence on the smoothness constant is worse than gradient descent. Liu et al. (2022) show that AdaGrad-Norm converges in the unconstrained setting when $f$ is quasi-convex, but their guarantee is worse than gradient descent. We remark that all AdaGrad-style algorithms mentioned above require tuning stepsizes, and are thus not parameter-free.

**Alternative justifications for normalization.** There are other justifications for why adaptive methods work outside of universality. Zhang et al. (2020a) study a generalized smoothness condition and show that in this setting tuned clipped gradient descent can outperform gradient descent. Because the effective stepsize used in clipped gradient descent is only a constant factor away from the effective stepsize in normalized gradient descent, (Zhang et al., 2020a), also show that this improvement holds for NGD. Zhang et al. (2020b) observe that gradient clipping and normalization methods outperform SGD when the stochastic gradient noise distribution is heavy-tailed. However, Kunstner et al. (2023) later observe that adaptive methods still do well even when the effect of the noise is limited.

## 3 Algorithms and theory

In this section we first review the different forms of adaptivity in gradient descent and normalized gradient descent, and then introduce our proposed algorithm DoWG. The roadmap for the rest of the paper is as follows: we first review the convergence of gradient descent in the Lipschitz and smooth settings, and highlight the problem of divergence under stepsize misspecification, and how normalization fixes that. Then, we introduce our main new algorithm, DoWG, and give our main theoretical guarantees for the algorithm. Finally, we evaluate the performance of DoWG on practical machine learning problems.

### 3.1 Baselines: gradient descent and normalized gradient descent

We start our investigation with the standard Gradient Descent (GD) algorithm:

$$x_{t+1} = \Pi_{\mathcal{X}}(x_t - \eta \nabla f(x_t)), \tag{GD}$$

where $\Pi_{\mathcal{X}}$ is the projection on $\mathcal{X}$ (when $\mathcal{X} = \mathbb{R}^d$, this is just the identity operator). The iterations (GD) require specifying the stepsize $\eta > 0$. When $f$ is $G$-Lipschitz, gradient descent achieves the following standard convergence guarantee:

**Theorem 1.** *Suppose that $f$ is convex with minimizer $x_*$. Let $f_* \stackrel{def}{=} f(x_*)$. Let $D_0 \stackrel{def}{=} \|x_0 - x_*\|$ be the initial distance to the optimum. Denote by $\hat{x}_T = \frac{1}{T}\sum_{t=0}^{T-1} x_t$ the average iterate returned by GD. Then:*

- *(Bubeck, 2015) If $f$ is $G$-Lipschitz, the average iterate satisfies for any stepsize $\eta > 0$:*

$$f(\bar{x}_T) - f_* \leq \frac{D_0^2}{\eta T} + \frac{\eta G^2}{2}, \tag{2}$$

- *(Nesterov, 2018) If $f$ is $L$-smooth, then for all $\eta < \frac{2}{L}$ the average iterate satisfies*

$$f(\hat{x}_T) - f_* \leq \frac{2LD_0^2}{4 + T\eta L(2 - L\eta)}. \tag{3}$$

Minimizing eq. (2) over $\eta$ gives $f(\bar{x}_T) - f_* = \mathcal{O}\left(\frac{D_0 G}{\sqrt{T}}\right)$ with $\eta = \frac{D_0}{G\sqrt{T}}$. We have several remarks to make about this rate for gradient descent. First, the optimal stepsize depends on both the distance

to the optimum $D_0$ and the Lipschitz constant $G$, and in fact, this rate is in general optimal (Nesterov, 2018, Theorem 3.2.1). Moreover, if we misspecify $D_0$ or $G$ while tuning $\eta$, this does not in general result in divergence but may result in a slower rate of convergence. On the other hand, for the smooth setting the optimal stepsize is $\eta = \frac{1}{L}$ for which $f(x_T) - f_* \leq \mathcal{O}\left(\frac{LD_0^2}{T}\right)$. Unfortunately, to obtain this rate we have to estimate the smoothness constant $L$ in order to choose a stepsize $\eta < \frac{2}{L}$, and this dependence is *hard*: if we overshoot the upper bound $\frac{2}{L}$, the iterations of gradient descent can diverge very quickly, as shown by Figure 1. Therefore, GD with a constant stepsize cannot be *universal*: we have to set the stepsize differently for smooth and nonsmooth objectives.

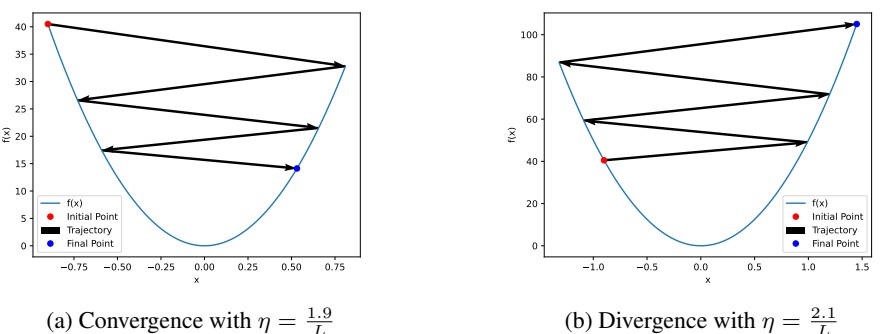

(a) Convergence with $\eta = \frac{1.9}{L}$        (b) Divergence with $\eta = \frac{2.1}{L}$

Figure 1: Two trajectories of gradient descent on the one-dimensional quadratic $f(x) = \frac{Lx^2}{2}$, with $L = 100$.

Normalized Gradient Descent (NGD) (Shor, 2012) consists of iterates of the form

$$x_{t+1} = \Pi_{\mathcal{X}}\left(x_t - \eta \frac{\nabla f(x_t)}{\|\nabla f(x_t)\|}\right). \tag{NGD}$$

The projection step above is not necessary, and the results for NGD also hold in the unconstrained setting where the projection on $\mathcal{X} = \mathbb{R}^d$ is just the identity. NGD has many benefits: it can escape saddle points that GD may take arbitrarily long times to escape (Murray et al., 2019), and can minimize functions that are quasi-convex and only locally Lipschitz (Hazan et al., 2015). One of the main benefits of normalized gradient descent is that normalization makes the method scale-free: multiplying $f$ by a constant factor $\alpha > 0$ and minimizing $\alpha f$ does not change the method's trajectory at all. This allows it to adapt to the Lipschitz constant $G$ in nonsmooth optimization as well as the smoothness constant $L$ for smooth objectives, as the following theorem states:

**Theorem 2.** *Under the same conditions as Theorem 1, the iterations generated by generated by (NGD) satisfy after $T$ steps satisfy:*

- *(Nesterov, 2018) If $f$ is $G$-Lipschitz, the minimal function suboptimality satisfies*

$$\min_{k \in \{0,1,\ldots,T-1\}} [f(x_k) - f_*] \leq G\left[\frac{D_0^2}{2\eta T} + \frac{\eta}{2}\right], \tag{4}$$

  *where $D_0 \stackrel{def}{=} \|x_0 - x_*\|$.*

- *(Levy, 2017; Grimmer, 2019) If $f$ is $L$-Lipschitz, the minimal function suboptimality satisfies*

$$\min_{k=0,\ldots,T-1} [f(x_k) - f_*] \leq \frac{L}{2}\left[\frac{D_0^2}{2\eta T} + \frac{\eta}{2}\right]^2. \tag{5}$$

Tuning eq. (4) in $\eta$ gives $\eta = \frac{D_0}{\sqrt{T}}$, and the stepsize is also optimal for eq. (5). This gives a convergence rate of $\frac{D_0 G}{\sqrt{T}}$ when $f$ is Lipschitz and $\frac{D_0^2 L}{T}$ when $f$ is smooth. Observe that NGD matches the dependence of gradient descent on $G$ and $L$ without any knowledge of it. Furthermore that, unlike GD where the optimal stepsize is $\frac{1}{L}$ in the smooth setting and $\frac{D_0}{G\sqrt{T}}$ in the nonsmooth

setting. The optimal stepsize for NGD is the same in both cases. Therefore, NGD is *universal*: the same method with the same stepsize adapts to nonsmooth and smooth objectives. Moreover, misspecification of the stepsize in NGD does not result in divergence, but just slower convergence. Another interesting property is that we only get a guarantee on the best iterate: this might be because NGD is non-monotonic, as Figure 2 (a) shows.

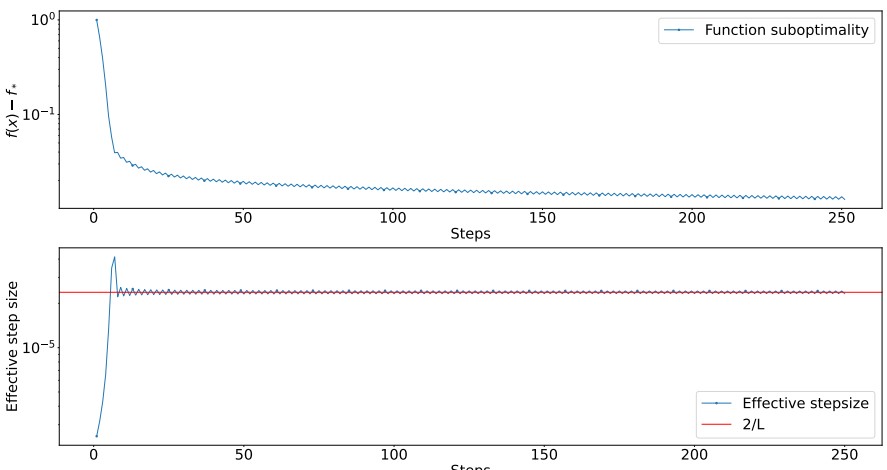

Figure 2: NGD iterations on $\ell_2$-regularized linear regression on the mushrooms dataset from Lib-SVM (Chang and Lin, 2011) with $\eta = 0.1$. Top (a) shows the function suboptimality over time. Observe that as the number of iterations grow, the method becomes non-monotonic. Bottom (b) shows the effective stepsize $\eta_{\text{eff},t} = \frac{0.1}{\|\nabla f(x_t)\|}$ over time.

**Edge of Stability Phenomena.** We may reinterpret NGD with stepsize $\eta$ as simply GD with a time-varying "effective stepsize" $\eta_{\text{eff},t} = \frac{\eta}{\|\nabla f(x_t)\|}$. We plot this effective stepsize for an $\ell_2$-regularized linear regression problem in Figure 2 (b). Observe that the stepsize sharply increases, then decreases until it starts oscillating around $\frac{2}{L}$. Recall that $\frac{2}{L}$ is the edge of stability for gradient descent: its iterates diverge when the stepsize crosses this threshold. Arora et al. (2022) observe this phenomenon for NGD, and give a detailed analysis of it under several technical assumptions and when the iterates are close to the manifold of local minimizers.

Theorem 2 offers an alternative, global, and less explicit explanation of this phenomenon: NGD matches the optimal gradient descent rate, and in order to do so it must drive the effective stepsize to be large. Specifically, suppose that we use the optimal stepsize $\eta = \frac{D_0}{\sqrt{T}}$, and call the best iterate returned by NGD $x_\tau$. Then $x_\tau$ satisfies $f(x_\tau) - f_* \leq \frac{D_0^2 L}{T}$ and therefore by smoothness

$$\|\nabla f(x_\tau)\| \leq \sqrt{2L\left(f(x_\tau) - f_*\right)} \leq \sqrt{\frac{L^2 D_0^2}{T}} = \frac{LD_0}{\sqrt{T}} = L\eta.$$

This implies $\eta_{\text{eff},\tau} = \frac{\eta}{\|\nabla f(x_t)\|} \geq \frac{1}{L}$. Therefore the effective stepsize at convergence is forced to grow to $\Omega\left(\frac{1}{L}\right)$. But if the effective stepsize increases too much and crosses the threshold $\frac{2}{L}$, the gradient norms start diverging, forcing the effective stepsize back down. Thus, NGD is *self-stabilizing*. We note that Edge of Stability phenomenon is not unique to NGD, and GD itself trains at the edge of stability for more complicated models where the smoothness also varies significantly over training (Cohen et al., 2021; Damian et al., 2023).

### 3.2 DoWG

We saw in the last section that NGD adapts to both the Lipschitz constant $G$ and the smoothness $L$, but we have to choose $\eta$ to vary with the distance to the optimum $D_0 = \|x_0 - x_*\|$. In this section, we develop a novel algorithm that adaptively estimates the distance to the optimum, and attains the optimal convergence rate of gradient descent for constrained convex and smooth optimization up to a logarithmic factor. Our algorithm builds upon the recently proposed Distance over Gradients (DoG)

algorithm developed by Ivgi et al. (2023). We call the new method DoWG (Distance over Weighted Gradients), and we describe it as Algorithm 1 below.

---

**Algorithm 1:** DoWG: Distance over Weighted Gradients

---

1 **Input**: initial point $x_0 \in \mathcal{X}$ Initial distance estimate $r_\epsilon > 0$.
2 **Initialize** $v_{-1} = 0$, $r_{-1} = r_\epsilon$.
3 **for** $t = 0, 1, 2, \ldots, T-1$ **do**
4      Update distance estimator: $\bar{r}_t \leftarrow \max\left(\|x_t - x_0\|,\ \bar{r}_{t-1}\right)$
5      Update weighted gradient sum: $v_t \leftarrow v_{t-1} + \bar{r}_t^2 \|\nabla f(x_t)\|^2$
6      Set the stepsize: $\eta_t \leftarrow \frac{\bar{r}_t^2}{\sqrt{v_t}}$
7      Gradient descent step: $x_{t+1} \leftarrow \Pi_{\mathcal{X}}(x_t - \eta_t \nabla f(x_t))$
8 **end**

---

DoWG uses the same idea of estimating the distance from the optimum by using the distance from the initial point as a surrogate, but instead of using the square root of the running gradient sum $G_t = \sum_{k=0}^{t} \|\nabla f(x_k)\|^2$ as the normalization, DoWG uses the square root of the weighted gradient sum $v_t = \sum_{k=0}^{t} \bar{r}_k^2 \|\nabla f(x_k)\|^2$. Observe that because the estimated distances $\bar{r}_t$ are monotonically increasing, later gradients have a larger impact on $v_t$ than earlier ones compared to $G_t$. Therefore, we may expect this to aid the method in adapting to the local properties of the problem once far away from the initialization $x_0$. We note that using a weighted sum of gradients is not new: AcceleGrad (Levy et al., 2018) uses time-varying polynomial weights and Adam (Kingma and Ba, 2015) uses exponentially decreasing weights. The difference is that DoWG chooses the weights adaptively based on the running distance from the initial point. This use of distance-based weighted averaging is new, and we are not aware of any previous methods that estimate the running gradient sum in this manner.

**Nonsmooth analysis.** The next theorem shows that DoWG adapts to the Lipschitz constant $G$ and the diameter $D$ of the set $\mathcal{X}$ if the function $f$ is nonsmooth but $G$-Lipschitz. We use the notation $\log_+ x = \log x + 1$ following (Ivgi et al., 2023).

**Theorem 3.** *(DoWG, Lipschitz $f$). Suppose that the function $f$ is convex, $G$-Lipschitz, and has a minimizer $x_* \in \mathcal{X}$. Suppose that the domain $\mathcal{X}$ is a closed convex set of (unknown) diameter $D > 0$. Let $r_\epsilon < D$. Then the output of Algorithm 1 satisfies for some $t \in \{0, 1, \ldots, T-1\}$*

$$f(\bar{x}_t) - f_* = \mathcal{O}\left[\frac{GD}{\sqrt{T}} \log_+ \frac{D}{r_\epsilon}\right],$$

*where $\bar{x}_t \overset{def}{=} \frac{1}{\sum_{k=0}^{t-1} \bar{r}_k^2} \sum_{k=0}^{t-1} \bar{r}_k^2 x_k$ is a weighted average of the iterates returned by the algorithm.*

**Discussion of convergence rate.** DoWG matches the optimal $\mathcal{O}\left(\frac{DG}{\sqrt{T}}\right)$ rate of tuned GD and tuned NGD up to an extra logarithmic factor. We note that the recently proposed algorithms DoG (Ivgi et al., 2023) and D-Adaptation (Defazio and Mishchenko, 2023) achieve a similar rate in this setting.

**Comparison with DoG.** As we discussed before, DoWG uses an adaptively weighted sum of gradients for normalization compared to the simple sum used by DoG. In addition, DoG uses the stepsize $\frac{\bar{r}_t}{\sqrt{\sum_{k=0}^{t} \|\nabla f(x_k)\|^2}}$, whereas the DoWG stepsize is pointwise larger: since $\bar{r}_k^2$ is monotonically increasing in $k$ we have

$$\eta_{\text{DoWG},t} = \frac{\bar{r}_t^2}{\sqrt{\sum_{k=0}^{t} \bar{r}_k^2 \|\nabla f(x_k)\|^2}} \geq \frac{\bar{r}_t^2}{\bar{r}_t \sqrt{\sum_{k=0}^{t} \|\nabla f(x_k)\|^2}} = \frac{\bar{r}_t}{\sqrt{\sum_{k=0}^{t} \|\nabla f(x_k)\|^2}} = \eta_{\text{DoG},t}.$$

Of course, the pointwise comparison may not reflect the practical performance of the algorithms, since after the first iteration the sequence of iterates $x_2, x_3, \ldots$ generated by the two algorithms can be very different. We observe in practice that DoWG is in general more aggressive, and uses larger stepsizes than both DoG and D-Adaptation (see Section 4).

**Smooth analysis.** Our next theorem shows that DoWG adapts to the smoothness constant and the diameter $D$ of the set $\mathcal{X}$.

**Theorem 4.** *(DoWG, Smooth f). Suppose that the function f is L-smooth, convex, and has a minimizer $x_* \in \mathcal{X}$. Suppose that the domain $\mathcal{X}$ is a closed convex set of diameter $D > 0$. Let $r_\epsilon < D$. Then the output of Algorithm 1 satisfies for some $t \in \{0, 1, \ldots, T-1\}$*

$$f(\bar{x}_t) - f_* = \mathcal{O}\left[\frac{LD^2}{T} \log_+ \frac{D}{r_\epsilon}\right],$$

*where $\bar{x}_t \stackrel{def}{=} \frac{1}{\sum_{k=0}^{t-1} \bar{r}_k^2} \sum_{k=0}^{t-1} \bar{r}_k^2 x_k$ is a weighted average of the iterates returned by the algorithm.*

The proof of this theorem and all subsequent results is relegated to the supplementary material. We note that the proof of Theorem 4 uses the same trick used to show the adaptivity of NGD to smoothness: we use the fact that $\|\nabla f(x)\| \leq \sqrt{2L(f(x) - f_*)}$ for all $x \in \mathcal{X}$ applied to a carefully-chosen weighted sum of gradients.

**Comparison with GD/NGD.** Both well-tuned gradient descent and normalized gradient descent achieve the convergence rate $\mathcal{O}\left(\frac{LD_0^2}{T}\right)$ where $D_0 = \|x_0 - x_*\| \leq D$ for the constrained convex minimization problem. Theorem 4 shows that DoWG essentially attains the same rate up to the difference between $D_0$ and $D$ and an extra logarithmic factor. In the worst case, if we initialize far from the optimum, we have $D_0 \simeq D$ and hence the difference is not significant. We note that DoG (Ivgi et al., 2023) suffers from a similar dependence on the diameter $D$ of $\mathcal{X}$, and can diverge in the unconstrained setting, where $\mathcal{X}$ is not compact. This can be alleviated by making the stepsize smaller by a polylogarithmic factor. A similar reduction of the stepsize also works for DoWG, and we provide the proof in Section 7 in the supplementary.

**Comparison with DoG.** After the initial version of this paper, Ivgi et al. (2023) reported a convergence guarantee for the *unweighted* average $\hat{x}_T = \frac{1}{T}\sum_{k=0}^{T-1} x_k$ returned by DoG. In particular, Proposition 3 in their work gives the rate

$$f(\hat{x}_T) - f_* = \mathcal{O}\left(\frac{L(D_0 \log_+ \frac{\bar{r}_T}{r_\epsilon} + \bar{r}_T)^2}{T}\right) = \mathcal{O}\left(\frac{LD^2}{T} \log_+^2 \frac{D}{\bar{r}_\epsilon}\right).$$

where $D_0 = \|x_0 - x_*\|$, and where in the second step we used the bound $D_0 \leq D$ and $\bar{r}_T \leq D$. This rate is the same as that achieved by the weighted average of the DoWG iterates up to an extra logarithmic factor $\log_+ \frac{D}{r_\epsilon}$. We note that DoG also has a guarantee in the stochastic setting, provided the gradients are bounded locally with a known constant, while in this work we have focused exclusively on the deterministic setting.

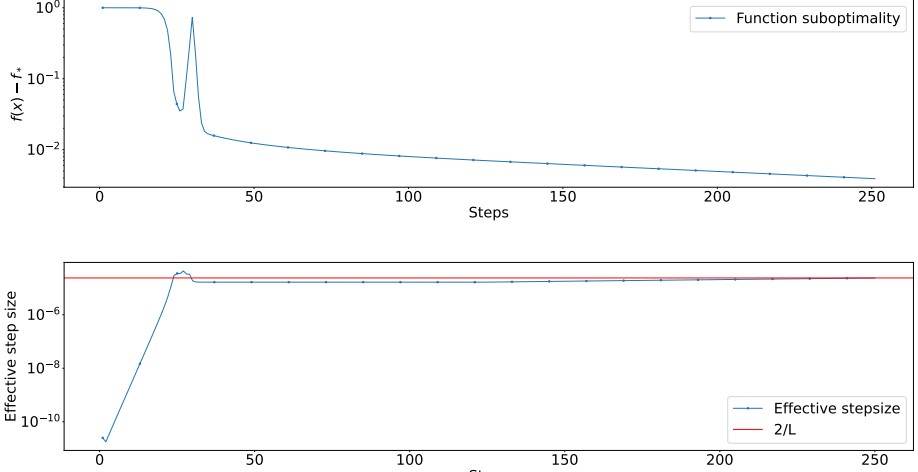

Figure 3: DoWG iterations on $\ell_2$-regularized linear regression on the mushrooms dataset from LibSVM (Chang and Lin, 2011) with $r_\epsilon = 10^{-6}$. Top (a) shows the function suboptimality over time. Observe that as the number of iterations grow, the method becomes non-monotonic. Bottom (b) shows the DoWG stepsize over time.

**Edge of Stability.** Like NGD, DoWG also tends to increase the stepsize and train at the edge of stability. The intuition from NGD carries over: in order to preserve the convergence rate of GD, DoWG tends to drive the stepsize larger. However, once it overshoots, the gradients quickly diverge, forcing the stepsize back down. Figure 3 shows the performance of DoWG and its stepsize on the same regularized linear regression problem as in Figure 2. Comparing the two figures, we observe that DoWG is also non-monotonic and trains close to the edge of stability, but its stepsize oscillates less than NGD's effective stepsize.

**Universality.** Theorems 4 and 3 together show that DoWG is universal, i.e. it almost recovers the convergence of gradient descent with tuned stepsizes in both the smooth and nonsmooth settings. As the optimal stepsize for gradient descent can differ significantly between the two settings, we believe that achieving both rates simultaneously without any parameter-tuning or search procedures is a significant strength of DoWG.

# 4 Experimental results

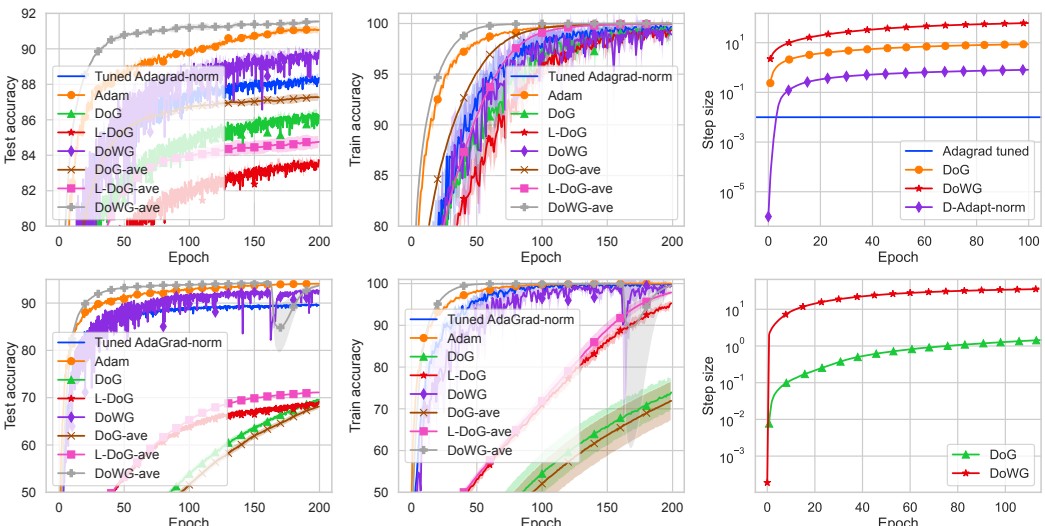

Figure 4: VGG11 (top) and ResNet-50 (bottom) training on CIFAR10. Left: test accuracy, middle: train loss, right: step sizes.

We compare DoWG to DoG, L-DoG from Ivgi et al. (2023), for all of which we also report performance of the polynomially-averaged iterate with power 8 as recommended by Ivgi et al. (2023). We also add comparison against Adam (Kingma and Ba, 2015) with cosine annealing and the standard step size $10^{-3}$. All methods are used with batch size 256 with no weight decay on a single RTX3090 GPU. We plot the results in Figure 4 with the results averaged over 8 random seeds. We train the VGG11 (Simonyan and Zisserman, 2015) and ResNet-50 (He et al., 2016) neural network architectures on CIFAR10 (Krizhevsky, 2009) using PyTorch (Paszke et al., 2019), and implement[1] DoWG on top of the DoG code[2]. Unsurprisingly, DoWG's estimates of the step size are larger than that of DoG and D-Adapt-norm, which also makes it less stable on ResNet-50. While the last iterate of DoWG gives worse test accuracy than Adam, the average iterate of DoWG often performs better.

Finally, we note that while both neural networks tested are generally nonsmooth, recent work shows *local* smoothness can significantly influence and be influenced by a method's trajectory (Cohen et al., 2022; Pan and Li, 2022). We believe this adaptivity to smoothness might explain the empirical difference between DoWG and DoG, but leave a rigorous discussion of adaptivity to local smoothness to future work.

---

[1] https://github.com/rka97/dowg
[2] https://github.com/formll/dog

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

# Supplementary material

## Contents

## 5 Algorithm-independent results

In this section we collect different results that are algorithm-independent, the first is a consequence of smoothness:

**Fact 1.** *Suppose that $f$ is smooth and lower bounded by $f_*$. Then for all $x \in \mathbb{R}^d$ we have,*

$$\|\nabla f(x)\|^2 \le 2L \left( f(x) - f_* \right).$$

*Proof.* This is a common result in the literature, and finds applications in convex and non-convex optimization see e.g. (Levy, 2017; Levy et al., 2018; Orabona and Cutkosky, 2020; Khaled and Richtárik, 2020). We include the proof for completeness. Let $x \in \mathbb{R}^d$ and define $x_+ = x - \frac{1}{L}\nabla f(x)$. Then by smoothness

$$f(x_+) \le f(x) + \langle \nabla f(x), x_+ - x \rangle + \frac{L}{2}\|x_+ - x\|^2$$

$$= f(x) - \frac{1}{L}\|\nabla f(x)\|^2 + \frac{1}{2L}\|\nabla f(x)\|^2$$

$$= f(x) - \frac{1}{2L}\|\nabla f(x)\|^2.$$

Because $f$ is lower bounded by $f_*$ we thus have

$$f_* \le f(x_+) \le f(x) - \frac{1}{2L}\|\nabla f(x)\|^2.$$

Rearranging gives $\|\nabla f(x)\|^2 \le 2L \left( f(x) - f_* \right)$. $\square$

The next two results are helpful algebraic identities that will be useful for the proof of DoWG.

**Lemma 1.** *(Ivgi et al., 2023, Lemma 4). Let $a_0, .., a_t$ be a nondecreasing sequence of nonnegative numbers. Then*

$$\sum_{k=1}^{t} \frac{a_k - a_{k-1}}{\sqrt{a_k}} \le 2 \left( \sqrt{a_t} - \sqrt{a_0} \right).$$

*Proof.* This is (Ivgi et al., 2023, Lemma 4). We include the proof for completeness:

$$\sum_{k=1}^{t} \frac{a_k - a_{k-1}}{\sqrt{a_k}} = \sum_{k=1}^{t} \frac{\left(\sqrt{a_k} - \sqrt{a_{k-1}}\right)\left(\sqrt{a_k} + \sqrt{a_{k-1}}\right)}{\sqrt{a_k}}$$

$$\leq 2 \sum_{k=1}^{t} \left(\sqrt{a_k} - \sqrt{a_{k-1}}\right)$$

$$= 2 \left(\sqrt{a_t} - \sqrt{a_0}\right).$$

□

**Lemma 2.** *((Ivgi et al., 2023, Lemma 3), similar to (Defazio and Mishchenko, 2023, Lemma 11)).*
*Let $s_0, s_1, \ldots, s_T$ be a positive increasing sequence. Then*

$$\max_{t \leq T} \sum_{i < t} \frac{s_i}{s_t} \geq \frac{1}{e} \left( \frac{T}{\log_+(s_T/s_0)} - 1 \right),$$

*where $\log_+ x \overset{def}{=} \log x + 1$.*

*Proof.* This is (Ivgi et al., 2023, Lemma 3). We include the proof for completeness: Define $K = \lceil \log \frac{s_T}{s_0} \rceil$ and $n = \lfloor \frac{T}{K} \rfloor$. Then,

$$\log \left( \frac{s_T}{s_0} \right) \geq \sum_{k=0}^{K-1} \log \left( \frac{s_{n(k+1)}}{s_{nk}} \right) \geq K \min_{k < K} \log \frac{s_{n(k+1)}}{s_{nk}}.$$

Rearranging and using $K = \lceil \log \frac{s_T}{s_0} \rceil$ gives

$$\min_{k < K} \log \frac{s_{n(k+1)}}{s_{nk}} \leq \frac{\log \frac{s_T}{s_0}}{K} \leq 1.$$

Therefore,

$$\min_{k < K} \frac{s_{n(k+1)}}{s_{nk}} \leq e.$$

Thus,

$$\max_{t \leq T} \sum_{i \leq t} \frac{s_i}{s_t} \geq \max_{t \leq T} n \frac{s_{t-n}}{s_t}$$

$$\geq \max_{k \leq K} n \frac{s_{n(k-1)}}{s_{nk}}$$

$$\geq e^{-1} n$$

$$= e^{-1} \left\lfloor \frac{T}{K} \right\rfloor$$

$$\geq e^{-1} \left( \frac{T}{K} - 1 \right)$$

$$\geq e^{-1} \left( \frac{T}{\log\left(\frac{s_T}{s_0}\right) + 1} - 1 \right).$$

□

## 6 Proofs for DoWG

This section collects proofs for DoWG. First, we give the following lemma, which holds under convexity alone (regardless of whether $f$ is smooth or Lipschitz).

**Lemma 3.** *Suppose that $f$ is convex and has minimizer $x_*$. For the iterations generated by Algorithm 1, we have*

$$\sum_{k=0}^{t-1} \bar{r}_k^2 \langle \nabla f(x_k), x_k - x_* \rangle \leq 2\bar{r}_t \left[ \bar{d}_t + \bar{r}_t \right] \sqrt{v_{t-1}}, \tag{6}$$

*where $\bar{d}_t = \max_{k \leq t} d_k$.*

*Proof.* This proof follows the proof of DoG (Ivgi et al., 2023, Lemma 1), itself a modification of the standard proof for adaptive cumulative gradient normalization methods (Gupta et al., 2017) incorporating insights from (Carmon and Hinder, 2022). We specifically modify the proof to handle the weighting scheme we use in DoWG. By the nonexpansivity of the projection we have

$$\begin{aligned}
d_{k+1}^2 &= \|x_{k+1} - x_*\|^2 \\
&\leq \|x_k - \eta_k \nabla f(x_k) - x_*\|^2 \\
&= \|x_k - x_*\|^2 - 2\eta_k \langle \nabla f(x_k), x_k - x_* \rangle + \eta_k^2 \|\nabla f(x_k)\|^2 \\
&= d_k^2 - 2\eta_k \langle \nabla f(x_k), x_k - x_* \rangle + \eta_k^2 \|\nabla f(x_k)\|^2.
\end{aligned}$$

Rearranging and dividing by $2\eta_k$ we get

$$\langle \nabla f(x_k), x_k - x_* \rangle \leq \frac{d_k^2 - d_{k+1}^2}{2\eta_k} + \frac{\eta_k}{2} \|\nabla f(x_k)\|^2.$$

Multiplying both sides by $\bar{r}_k^2$ we get

$$\bar{r}_k^2 \langle \nabla f(x_k), x_k - x_* \rangle \leq \frac{1}{2} \frac{\bar{r}_k^2}{\eta_k} \left[ d_k^2 - d_{k+1}^2 \right] + \frac{1}{2} \bar{r}_k^2 \eta_k \|\nabla f(x_k)\|^2.$$

Summing up as $k$ varies from $0$ to $t-1$ we get

$$\sum_{k=0}^{t-1} \bar{r}_k^2 \langle \nabla f(x_k), x_k - x_* \rangle \leq \frac{1}{2} \underbrace{\left[ \sum_{k=0}^{t-1} \frac{\bar{r}_k^2}{\eta_k} \left( d_k^2 - d_{k+1}^2 \right) \right]}_{(A)} + \frac{1}{2} \underbrace{\left[ \sum_{k=0}^{t-1} \bar{r}_k^2 \eta_k \|\nabla f(x_k)\|^2 \right]}_{(B)}. \tag{7}$$

We shall now bound each of the terms (A) and (B). We have

$$\begin{aligned}
(A) &= \sum_{k=0}^{t-1} \frac{\bar{r}_k^2}{\eta_k} \left( d_k^2 - d_{k+1}^2 \right) \\
&= \sum_{k=0}^{t-1} \sqrt{v_k} \left( d_k^2 - d_{k+1}^2 \right) \tag{8} \\
&= d_0^2 \sqrt{v_0} - d_t^2 \sqrt{v_{t-1}} + \sum_{k=1}^{t-1} d_k^2 \left( \sqrt{v_k} - \sqrt{v_{k-1}} \right) \tag{9} \\
&\leq \bar{d}_t^2 \sqrt{v_0} - d_t^2 \sqrt{v_{t-1}} + \bar{d}_t^2 \sum_{k=1}^{t-1} \left( \sqrt{v_k} - \sqrt{v_{k-1}} \right) \tag{10} \\
&= \sqrt{v_{t-1}} \left[ \bar{d}_t^2 - d_t^2 \right] \tag{11} \\
&\leq 4\bar{r}_t \bar{d}_t \sqrt{v_{t-1}}, \tag{12}
\end{aligned}$$

where eq. (8) holds by definition of the DoWG stepsize $\eta_k$, eq. (9) holds by telescoping, eq. (10) holds because $v_k = v_{k-1} + \bar{r}_k^2 \|\nabla f(x_k)\|^2 \geq v_{k-1}$ and hence $\sqrt{v_k} \geq \sqrt{v_{k-1}}$, and $d_k^2 \leq \bar{d}_t^2$ by definition. Equation (11) just follows by telescoping. Finally observe that $\bar{d}_t^2 - d_t^2 = d_s^2 - d_t^2$ for some $s \in [t]$, and $d_s^2 - d_t^2 = (d_s - d_t)(d_s + d_t)$. Then by the triangle inequality and that the sequence

$\bar{r}_k$ is monotonically nondecreasing we have

$$
\begin{aligned}
d_s - d_t = \|x_s - x_*\| - \|x_t - x_*\| \\
\leq \|x_s - x_t\| \\
\leq \|x_s - x_0\| + \|x_t - x_0\| \\
= r_s + r_t \\
\leq \bar{r}_s + \bar{r}_t \\
\leq 2\bar{r}_t.
\end{aligned}
$$

Therefore $d_s^2 - d_t^2 \leq (\bar{r}_s + \bar{r}_t)(d_s + d_t) \leq 4\bar{r}_t\bar{d}_t$. This explains eq. (12).

For the second term in eq. (7), we have

$$
\begin{aligned}
(\text{B}) &= \sum_{k=0}^{t-1} \bar{r}_k^2 \eta_k \|\nabla f(x_k)\|^2 \\
&= \sum_{k=1}^{t-1} \frac{\bar{r}_k^4}{\sqrt{v_k}} \|\nabla f(x_k)\|^2 \\
&= r_0^2 \sqrt{v_0} + \sum_{k=1}^{t-1} \frac{\bar{r}_k^4}{\sqrt{v_k}} \|\nabla f(x_k)\|^2 \\
&\leq \bar{r}_t^2 \sqrt{v_0} + \bar{r}_t^2 \sum_{k=1}^{t-1} \frac{\bar{r}_k^2 \|\nabla f(x_k)\|^2}{\sqrt{v_k}} \\
&= \bar{r}_t^2 \sqrt{v_0} + \bar{r}_t^2 \sum_{k=1}^{t-1} \frac{v_k - v_{k-1}}{\sqrt{v_k}} \\
&= \bar{r}_t^2 \sqrt{v_0} + \bar{r}_t^2 \sum_{k=1}^{t-1} \frac{v_k - v_{k-1}}{\sqrt{v_k}} \\
&\leq \bar{r}_t^2 \sqrt{v_0} + 2\bar{r}_t^2 \left[ \sqrt{v_{t-1}} - \sqrt{v_0} \right] \qquad (13) \\
&= 2\bar{r}_t^2 \sqrt{v_{t-1}}. \qquad (14)
\end{aligned}
$$

where eq. (13) is by Lemma 1. Plugging eqs. (12) and (14) in eq. (7) gives

$$
\begin{aligned}
\sum_{k=0}^{t-1} \bar{r}_k^2 \langle \nabla f(x_k), x_k - x_* \rangle &\leq 2\bar{r}_t \bar{d}_t \sqrt{v_{t-1}} + \bar{r}_t^2 \sqrt{v_{t-1}} \\
&\leq 2\bar{r}_t \left[ \bar{d}_t + \bar{r}_t \right] \sqrt{v_{t-1}}.
\end{aligned}
$$

$\square$

## 6.1  Smooth case

We now prove the convergence of DoWG under smoothness. In particular, we shall use Fact 1 and the DoWG design to bound the weighted cumulative error $S_t = \sum_{k=0}^{t-1} \bar{r}_k^2 [f(x_k) - f_*]$ by its square root multiplied by a problem-dependent constant. We note that a similar trick is used in the analysis of AdaGrad-Norm (Levy et al., 2018), in reductions from online convex optimization to stochastic smooth optimization (Orabona and Cutkosky, 2020), and in the method of (Carmon and Hinder, 2022). However, in all the mentioned cases, the *unweighted* error $M_t = \sum_{k=0}^{t-1} [f(x_k) - f_*]$ is bounded by its square root. Here, DoWG's design allows us to bound the weighted errors $S_t$ instead.

*Proof of Theorem 4.* We start with Lemma 3. Let $t \in [T]$. By eq. (6) we have

$$
\sum_{k=0}^{t-1} \bar{r}_k^2 \langle \nabla f(x_k), x_k - x_* \rangle \leq 2\bar{r}_t \left[ \bar{d}_t + \bar{r}_t \right] \sqrt{v_{t-1}}. \qquad (15)
$$

Observe that by convexity we have

$$\langle \nabla f(x_k), x_k - x_* \rangle \geq f(x_k) - f_*.$$

Using this to lower bound the left-hand side of eq. (15) gives

$$\sum_{k=0}^{t-1} \bar{r}_k^2 \left[ f(x_k) - f_* \right] \leq \sum_{k=0}^{t-1} \bar{r}_k^2 \left\langle \nabla f(x_k), x_k - x_* \right\rangle$$
$$\leq 2\bar{r}_t \left[ \bar{d}_t + \bar{r}_t \right] \sqrt{v_{t-1}}. \tag{16}$$

We have by smoothness that $\|\nabla f(x)\|^2 \leq 2L(f(x) - f_*)$ for all $x \in \mathbb{R}^d$, therefore

$$v_{t-1} = \sum_{k=0}^{t-1} \bar{r}_k^2 \|\nabla f(x_k)\|^2 \leq 2L \sum_{k=0}^{t-1} \bar{r}_k^2 \left[ f(x_k) - f_* \right].$$

Taking square roots we get

$$\sqrt{v_{t-1}} \leq \sqrt{2L} \sqrt{\sum_{k=0}^{t-1} \bar{r}_k^2 \left[ f(x_k) - f_* \right]}. \tag{17}$$

Using eq. (17) in eq. (16) gives

$$\sum_{k=0}^{t-1} \bar{r}_k^2 \left[ f(x_k) - f_* \right] \leq 2\sqrt{2L} \bar{r}_t \left( \bar{d}_t + \bar{r}_t \right) \sqrt{\sum_{k=0}^{t-1} \bar{r}_k^2 \left[ f(x_k) - f_* \right]}.$$

If $f(x_k) - f_* = 0$ for some $k \in [t-1]$ then the statement of the theorem is trivial. Otherwise, we can divide both sides by the latter square root to get

$$\sqrt{\sum_{k=0}^{t-1} \bar{r}_k^2 \left[ f(x_k) - f_* \right]} \leq 2\sqrt{2L} \bar{r}_t \left( \bar{d}_t + \bar{r}_t \right).$$

Squaring both sides gives

$$\sum_{k=0}^{t-1} \bar{r}_k^2 \left[ f(x_k) - f_* \right] \leq 8L \bar{r}_t^2 \left( \bar{d}_t + \bar{r}_t \right)^2.$$

Dividing both sides by $\sum_{k=0}^{t-1} \bar{r}_k^2$ we get

$$\frac{1}{\sum_{k=0}^{t-1} \bar{r}_k^2} \sum_{k=0}^{t-1} \bar{r}_k^2 \left[ f(x_k) - f_* \right] \leq \frac{8L \bar{r}_t^2 \left( \bar{d}_t + \bar{r}_t \right)^2}{\sum_{k=0}^{t-1} \bar{r}_k^2}$$
$$= \frac{8L \left( \bar{d}_t + \bar{r}_t \right)^2}{\sum_{k=0}^{t-1} \frac{\bar{r}_k^2}{\bar{r}_t^2}}.$$

By convexity we have

$$f(\bar{x}_t) - f_* \leq \frac{1}{\sum_{k=0}^{t-1} \bar{r}_k^2} \sum_{k=0}^{t-1} \bar{r}_k^2 \left[ f(x_k) - f_* \right]$$
$$\leq \frac{8L \left( \bar{d}_t + \bar{r}_t \right)^2}{\sum_{k=0}^{t-1} \frac{\bar{r}_k^2}{\bar{r}_t^2}}. \tag{18}$$

By Lemma 2 applied to the sequence $s_k = \bar{r}_k^2$ we have that for some $t \in [T]$

$$\sum_{k=0}^{t-1} \frac{\bar{r}_k^2}{\bar{r}_t^2} \geq \frac{1}{e} \left( \frac{T}{\log_+ \frac{\bar{r}_T^2}{\bar{r}_0^2}} - 1 \right).$$

Because $\mathcal{X}$ has diameter $D$ we have $\bar{r}_T^2 \leq D^2$ and therefore

$$\sum_{k=0}^{t-1} \frac{\bar{r}_k^2}{\bar{r}_t^2} \geq \frac{1}{e}\left(\frac{T}{\log_+ \frac{D^2}{\bar{r}_0^2}} - 1\right). \tag{19}$$

We now have two cases:

- If $T \geq 2\log_+ \frac{D^2}{\bar{r}_0^2}$ then $\frac{T}{\log \frac{D^2}{\bar{r}_0^2}} - 1 \geq \frac{T}{2\log \frac{D^2}{\bar{r}_0^2}}$ and we use this in eqs. (18) and (19) to get

$$f(\bar{x}_t) - f_* \leq \frac{16eL\left(\bar{d}_t + \bar{r}_t\right)^2}{T} \log \frac{\bar{r}_T^2}{\bar{r}_0^2}.$$

  Observe that because $\mathcal{X}$ has diameter at most $D$ we have $\bar{d}_t + \bar{r}_t \leq 2D$, therefore

$$f(\bar{x}_t) - f_* \leq \frac{64eLD^2}{T} \log_+ \frac{\bar{r}_T^2}{\bar{r}_0^2} = \mathcal{O}\left[\frac{LD^2}{T} \log_+ \frac{D}{\bar{r}_0}\right].$$

- If $T < 2\log_+ \frac{D^2}{\bar{r}_0^2}$, then $1 < \frac{2\log_+ \frac{D^2}{\bar{r}_0^2}}{T}$. Let $t \in [T]$. Using smoothness and this fact we have

$$f(\bar{x}_t) - f_* \leq \frac{L}{2}\|\bar{x}_t - x_*\|^2 \leq \frac{L\|\bar{x}_t - x_*\|^2}{T} \log_+ \frac{D^2}{\bar{r}_0^2}.$$

  Observe $\bar{x}_t, x_* \in \mathcal{X}$ and $\mathcal{X}$ has diameter $D$, hence $\|\bar{x}_t - x_*\|^2 \leq D^2$ and we get

$$f(\bar{x}_t) - f_* \leq \frac{LD^2}{T} \log_+ \frac{D^2}{\bar{r}_0^2} = \mathcal{O}\left(\frac{LD^2}{T} \log_+ \frac{D}{\bar{r}_0}\right).$$

Thus in both cases we have that $f(\bar{x}_t) - f_* = \mathcal{O}\left(\frac{LD^2}{T} \log_+ \frac{D}{\bar{r}_0}\right)$, this completes our proof. $\quad\square$

## 6.2 Nonsmooth case

We now give the proof of DoWG's convergence when $f$ is Lipschitz.

*Proof of Theorem 3.* We start with Lemma 3. Let $t \in [T]$. By eq. (6) we have

$$\sum_{k=0}^{t-1} \bar{r}_k^2 \langle \nabla f(x_k), x_k - x_* \rangle \leq 2\bar{r}_t \left[\bar{d}_t + \bar{r}_t\right] \sqrt{v_{t-1}}. \tag{20}$$

Observe that by convexity we have

$$\langle \nabla f(x_k), x_k - x_* \rangle \geq f(x_k) - f_*.$$

Using this to lower bound the left-hand side of eq. (20) gives

$$\sum_{k=0}^{t-1} \bar{r}_k^2 [f(x_k) - f_*] \leq \sum_{k=0}^{t-1} \bar{r}_k^2 \langle \nabla f(x_k), x_k - x_* \rangle$$
$$\leq 2\bar{r}_t \left[\bar{d}_t + \bar{r}_t\right] \sqrt{v_{t-1}}. \tag{21}$$

We have by the fact that $f$ is $G$-Lipschitz that $\|\nabla f(x)\|^2 \leq G^2$ for all $x \in \mathcal{X}$. Therefore,

$$v_{t-1} = \sum_{k=0}^{t-1} \bar{r}_k^2 \|\nabla f(x_k)\|^2$$
$$\leq \bar{r}_t^2 \sum_{k=0}^{t-1} \|\nabla f(x_k)\|^2$$
$$\leq \bar{r}_t^2 G^2 T.$$

Taking square roots and plugging into eq. (21) gives

$$\sum_{k=0}^{t-1} \bar{r}_k^2 \left[f(x_k) - f_*\right] \leq 2\bar{r}_t^2 \left[\bar{d}_t + \bar{r}_t\right] G\sqrt{T}.$$

Dividing both sides by $\sum_{k=0}^{t-1} \bar{r}_k^2$ we get

$$\frac{1}{\sum_{k=0}^{t-1} \bar{r}_k^2} \sum_{k=0}^{t-1} \bar{r}_k^2 \left[f(x_k) - f_*\right] \leq \frac{2\left[\bar{d}_t + \bar{r}_t\right] G\sqrt{T}}{\sum_{k=0}^{t-1} \frac{\bar{r}_k^2}{\bar{r}_t^2}}. \tag{22}$$

By Lemma 2 applied to the sequence $s_k = \bar{r}_k^2$ we have that for some $t \in [T]$

$$\sum_{k=0}^{t-1} \frac{\bar{r}_k^2}{\bar{r}_t^2} \geq \frac{1}{e} \left(\frac{T}{\log_+ \frac{\bar{r}_T^2}{\bar{r}_0^2}} - 1\right).$$

Because $\bar{r}_T \leq D$ we further have

$$\sum_{k=0}^{t-1} \frac{\bar{r}_k^2}{\bar{r}_t^2} \geq \frac{1}{e} \left(\frac{T}{\log_+ \frac{D^2}{\bar{r}_0^2}} - 1\right). \tag{23}$$

We now have two cases:

- If $T \geq 2\log_+ \frac{D^2}{\bar{r}_0^2}$: then $\frac{T}{\log \frac{\bar{r}_T^2}{\bar{r}_0^2}} - 1 \geq \frac{T}{2\log \frac{\bar{r}_T^2}{\bar{r}_0^2}}$. We can use this in eq. (22) alongside eq. (23) and

  the fact that $\log_+ x^2 = \max(\log x^2, 1) = \max(2\log x, 1) \leq 2\log_+ x$ to get

  $$\frac{1}{\sum_{k=0}^{t-1} \bar{r}_k^2} \sum_{k=0}^{t-1} \bar{r}_k^2 \left[f(x_k) - f_*\right] \leq \frac{8\left[\bar{d}_t + \bar{r}_t\right] G}{\sqrt{T}} \log_+ \frac{D}{\bar{r}_0}.$$

  Because the diameter of $\mathcal{X}$ is bounded by $D$ we have $\bar{r}_T \leq D$ and $\bar{d}_t \leq D$, using this and convexity we get

  $$
  \begin{aligned}
  f(\bar{x}_t) - f_* &\leq \frac{1}{\sum_{k=0}^{t-1} \bar{r}_k^2} \sum_{k=0}^{t-1} \bar{r}_k^2 \left[f(x_k) - f_*\right] \\
  &\leq \frac{8\left[\bar{d}_t + \bar{r}_t\right] G}{\sqrt{T}} \log \frac{\bar{r}_T}{\bar{r}_0} \\
  &\leq \frac{16DG}{\sqrt{T}} \log \frac{D}{r_0}.
  \end{aligned}
  $$

- If $T < 2\log_+ \frac{D^2}{\bar{r}_0^2}$: then

  $$1 < \frac{2\log_+ \frac{D^2}{\bar{r}_0^2}}{T} \leq \frac{4\log_+ \frac{D}{\bar{r}_0}}{T}. \tag{24}$$

  By convexity and Cauchy-Schwartz we have

  $$
  \begin{aligned}
  f(\bar{x}_t) - f_* &\leq \langle \nabla f(\bar{x}_t), \bar{x}_t - x_*\rangle \\
  &\leq \|\nabla f(\bar{x}_t)\| \|\bar{x}_t - x_*\|. 
  \end{aligned} \tag{25}
  $$

  Because $f$ is $G$-Lipschitz then $\|\nabla f(\bar{x}_t)\| \leq G$ and because $\mathcal{X}$ has diameter $D$ we have $\|\bar{x}_t - x_*\| \leq D$. Using this and eq. (24) in eq. (25) gives

  $$
  \begin{aligned}
  f(\bar{x}_t) - f_* &\leq DG \\
  &< \frac{4DG\log_+ \frac{D}{\bar{r}_0}}{T}.
  \end{aligned}
  $$

  Now because $T \geq 1$ we have $\sqrt{T} \leq T$ and hence

  $$f(\bar{x}_t) - f_* \leq \frac{4DG\log_+ \frac{D}{\bar{r}_0}}{\sqrt{T}}.$$

In both cases, we have that $f(\bar{x}_t) - f_* = \mathcal{O}\left(\frac{4DG\log_+ \frac{D}{\bar{r}_0}}{\sqrt{T}}\right)$, and this completes our proof. $\qquad \square$

# 7 Unconstrained domain extension

In this section we consider the case where the domain set is unbounded, and we seek dependence only on $d_0 = \|x_0 - x_*\|$. We use the same technique for handling the unconstrained problem as (Ivgi et al., 2023) in this section and consider DoWG iterates with the reduced stepsizes

$$\eta_t = \frac{\bar{r}_t^2}{\sqrt{v_t} \log \frac{2v_t}{v_0}} \qquad\qquad v_t = v_{t-1} + \bar{r}_t^2 \|\nabla f(x_t)\|^2. \qquad (26)$$

We prove that with this stepsize, the iterates do not venture far from the initialization. The proof follows (Ivgi et al., 2023).

**Lemma 4.** *(Stability). For the iterates $x_{t+1} = x_t - \eta_t \nabla f(x_t)$ following the stepsize scheme given by (26) we have $\bar{d}_t^2 \le 12 d_0$ and $\bar{r}_t^2 \le 32 d_0^2$ provided that $r_0 \le d_0$.*

*Proof.* By expanding the square and convexity

$$d_{k+1}^2 - d_k^2 \le \eta_t^2 \|g_t\|^2.$$

Summing up as $t$ varies from $k = 1$ to $k = t$ and using (Ivgi et al., 2023, Lemma 6)

$$d_t^2 - d_1^2 \le \sum_{k=1}^t \frac{\bar{r}_k^4}{v_k} \frac{\|\nabla f(x_k)\|^2}{4 \log^2 \frac{2v_k}{v_0}} \le \frac{\bar{r}_t^2}{4} \sum_{k=1}^t \frac{v_k - v_{k-1}}{v_k \log_+^2 \frac{v_k}{v_0}} \le \frac{\bar{r}_t^2}{4}.$$

Therefore we have $d_t^2 \le d_1^2 + \frac{\bar{r}_t^2}{4}$. Now suppose for the sake of induction that $\bar{r}_t^2 \le 8 d_1^2$, then applying the last equation we get $d_t^2 \le 3 d_1^2$. Taking square roots gives $d_t \le \sqrt{3} d_1$. By the triangle inequality we then get

$$\|x_{t+1} - x_0\| \le \|x_{t+1} - x_*\| + \|x_* - x_0\| \le (1 + \sqrt{3}) d_1.$$

Squaring both sides gives $\|x_{t+1} - x_0\|^2 \le (1 + \sqrt{3})^2 d_1^2 \le 8 d_1^2$. This completes our induction and we have $\bar{r}_t^2 \le 8 d_1^2$ for all $t$. Finally, observe that

$$d_1 = \|x_1 - x_*\| \le \|x_1 - x_0\| + \|x_0 - x_*\| = r_\epsilon + d_0 \le 2 d_0.$$

It follows that $\bar{r}_t^2 \le 8 d_1^2 \le 32 d_0^2$ for all $t$. Finally, we have $d_t^2 \le d_1^2 + \frac{\bar{r}_t^2}{4} \le 3 d_1^2 \le 12 d_0^2$. This completes our proof. $\qquad\square$

Therefore the iterates stay bounded. The rest of the proof then follows Theorems 3 and 4 and is omitted for simplicity.. In both cases it gives the same results with $D_0 = \|x_0 - x_*\|$ instead of $D$, up to extra constants and polylogarithmic factors.

