# OpenReview forum: "DoWG Unleashed: An Efficient Universal Parameter-Free Gradient Descent Method"
_NeurIPS.cc/2023/Conference — NeurIPS 2023 poster_

### Official Review · Reviewer_jMMU · 2023-07-11

**Soundness:** 4 excellent
**Presentation:** 3 good
**Contribution:** 2 fair
**Rating:** 6
**Confidence:** 1

**Summary:**

This paper is concerned with parameter free - adaptive first order optimisation method, that is a method that achieves optimal rates of convergence in the class of function considered without having access to a priori quantities such as smoothness of the function or the minimum value of the function.
The authors discuss on normalized gradient descent and they show that this method is adaptive to the smoothness of the function when the step-size is proportional to the distance to the minimizer.
The main contribution of the paper is to introduce a new estimation of the distance to the minimizer that achieves optimal rate up to logarithm factor. The method is an improvement over the so-called distance over gradients (DoW) by introducing weights (DoWG) so that it gives more weight to the last gradients.
Numerical experiments are provided, including comparison to Adam (which is not in the same category...).

**Strengths:**

I am not a specialist of the field so my contribution in this review was to check the proofs, also in the supplementary material.

All the proofs are correct.

The paper is globally nicely written, easy to read and accessible for an outsider of the field as I am.

Although from the theoretical point of view, DoWG is incremental (correct me if I am wrong), numerical experiments tend to show that it is clearly more efficient practically.

Although the resulting algorithm performs poorly compared to Adam, it seems a fair contribution on the understanding and the technics in the class of methods considered.


**Weaknesses:**

This work appears as an improvement over DoG. It is a bit surprising that the result on Normalized Gradient Descent is new; I was not able to find it in the literature although, again, I am not an expert in the field.
The proposed method DoWG of using weights in front of the gradients is a natural idea.
From the theoretical results, DoWG seems a rather incremental advance over DoG.

The resulting algorithm is probably not going to have any impact on the optimization method in deep learning.


**Questions:**

Could the rate obtained in the analysis of NGD be improved?

- line 131: gives eta so that
- line 554: Therefore.



**Limitations:**

do no apply.

---

> ### Author Rebuttal · Authors · 2023-08-07
>
> Thank you so much for your evaluation of our work.
>
> 1. "The resulting algorithm is probably not going to have any impact on the optimization method in deep learning." On the contrary, the method has been independently implemented in at least one GitHub repository with 500+ stars for fine-tuning stable diffusion models, which we became aware of only after the submssion. We cannot link the repository per NeurIPS rules, but we can provide it to the Program Chairs for verification.
> 2. On improving the rates for NGD: The rate for NGD in the non-smooth setting matches the optimal rate $\frac{GD}{\sqrt{T}}$ for first-order algorithms. In the smooth setting, we do not know if it can be improved. Our theory and experiments suggest the agreement is pretty tight, as we observe the effective stepsize indeed oscillates around $1/L$.

---

> > ### Comment · Reviewer_jMMU · 2023-08-14
> >
> > Thank you for your answer.

---

### Official Review · Reviewer_oAov · 2023-07-11

**Soundness:** 2 fair
**Presentation:** 3 good
**Contribution:** 2 fair
**Rating:** 6
**Confidence:** 4

**Summary:**

The paper presents a modification to the recently proposed DoG algorithm to obtain a adaptive algorithm for the deterministic, convex and [Lipschitz or Smooth] setting that does not need any hyperparameter to achieve a convergence rate competitive with algorithms that know the problem-specific constants.

The submission discusses the benefits of normalization, linking its behavior to the edge of stability phenomenon and providing an analysis that normalized gradient descent obtains the convergence of GD on smooth function with step-size 1/L without needing to know the smoothness constant.

**Strengths:**

The submission is well presented and I could understand the main objective of the paper at first read. The DoWG algorithm, and especially its proof, is simpler than the one found the prior work of Ivgi et al. (2023) on DoG, and would be an easier starting point for readers trying to understand algorithms that can adapt to the initial distance to the optimum.

**Weaknesses:**

Beyond the DoWG algorithm, the contributions are also already known in the literature. To the credit of the submission, this is clearly mentioned for the observation that normalized gradient descent on smooth functions leads to a similar behavior to the edge of stability phenomenon (Arora et al., 2022). However, the result that normalized gradient descent obtains the smooth rate can be found in the work of Levy (2017), which seems to have been missed.

The submission also suffers from a lack of formalism on some of its claims. Some concepts such as weak and strong adaptivity, “adapting to geometry” and what is meant by the edge of stability are left vague, and would benefit from a formal definition and nuanced discussion.

If those issues, detailed in the question section, are addressed in a revision, I will increase my score to 6.

**Questions:**

**Prior work**

I found the discussion of the adaptivity to smoothness of normalized GD and the key inequalities (5—7) useful to understand how DoWG achieves a similar result. However, the paragraph prior to Theorem 4 should cite Levy (2017). Their analysis of what they call the AdaNGDk algorithm (an overly general form that recovers normalized gradient descent with a decreasing step-size of the order $1/\sqrt{t}$
for k=1), in Theorem 2.1 is the equivalent of theorem 4 in the current submission, except with a decreasing step-size of $1/\sqrt{t}$ rather than the constant but horizon-dependent step-size $1/\sqrt{T}$ used here.

Kfir Levy, “Online to Offline Conversions, Universality and Adaptive Minibatch Sizes”, NeurIPS 2017

---

**Formal definition for some concepts**

**Strong vs. Weak Adaptivity**

The working definitions of “weak” and “strong” adaptivity are not clear from the current writing, or seems to lead to undesirable conclusions. I find the separation useful, and being able to formalize how sub-optimal the “adaptivity” of AdaGrad-Norm is in the smooth setting would be helpful.

However, the definition of weakly adaptive requires that “[weak-adaptivity] just seeks non-divergence given stepsize misspecification. […] the algorithm’s objective is to ensure that the learning process does not result in divergence, even if the chosen stepsize is not optimal” (L145). This definition is problematic because an algorithm that does not move, or gradient descent with a step-size of 0, satisfies the above definition.

Similarly, for strongly adaptive, the text requires that “an algorithm [is strongly adaptive if] it preserves the convergence rate of optimally tuned gradient descent without any manual tuning.” The proposed DoWG algorithm does not fit this description as the rate worse, if only by a log-factor.

A formalization of the above that could work would be to say that an algorithm is strongly adaptive in a problem setting if it achieves the same convergence rate, up to polylogarithmic terms, as an algorithm that knows problem specific constants. Weakly-adaptive could similarly be defined by allowing for multiplicative polynomial factors.

**Adapting to geometry**

The term “adapting to geometry” is used in multiple places, but it isn’t clear what is meant by that statement. This term is often used in the literature as a way to convey the intuition of why a method is good, but with limited formalism. For example, Newton adapts to the geometry of the problem by using the Hessian, or AdaGrad adapts to the geometry of the problem by finding a preconditioner that is competitive with the optimal one in hindsight. On each usage of “adapting to geometry”, I do not understand what is meant,. I strongly suggest avoiding it and using a more direct description instead.

- L32: “We say an algorithm is universal if it adapts to many different problem geometries or regularity conditions on the function f”
(should be removed and only mention regularity condition)
- L56: “In particular, such adaptive methods empirically show the ability to adapt to the local geometry of the problem (Cohen et al., 2022), especially compared to plain (stochastic) gradient descent (Pan and Li, 2022).”
L107: “There are other justifications for why adaptive methods work outside of adapting to geometry.”
- L246 “Therefore, we may may expect this to aid the method in adapting to the geometry of the problem once far away from the initialization x0.”

---

**A mention of the difficulties of obtaining similar results in online learning would be helpful to the reader.**

While the online-learning algorithms of Streeter and McMahan (2012), Orabona et Pal (2016) and Orabona and Cutkosky (2020) aim for a similar goal of not having to know the diameter of the set and the Lipschitz constant, the rates presented for the current algorithm or those in the prior works of of Ivgi et al. (2023) and Carmon and Hinder (2022) are not achievable in the adversarial that is common in online learning, see Cutkosky and Boahen (2016, 2017).

Ashok Cutkosky, Kwabena Boahen, “Online Convex Optimization with Unconstrained Domains and Losses”, NeurIPS 2016
Ashok Cutkosky, Kwabena Boahen, “Online Learning Without Prior Information”, COLT 2017

---

**Question on Universality and the edge of stability**

The convergence of normalized GD or DoWG are not discussed under strong-convexity, which I think should be mentioned given the focus on universal algorithms. Especially as some alternative algorithms such as the Polyak step-size do benefit from strong-convexity, whether in the smooth or Lipschitz+bounded set case.

Given that the empirical results shown in Figure 2 are on a strongly convex problem and the effective step-size oscillates around a constant 2/L, I am interpreting it as the algorithm *not* achieving a linear rate (as the gradient norm should also go down as a linear rate, which does not seem to be the case if $\eta/\Vert\nabla f(x_t)\Vert$ is constant)?


---

**typo?**
- L80: "Orvieto et al. (2022) show that a variant of the Polyak stepsize with decreasing stepsizes can recover the convergence rate of gradient descent in the deterministic setting, provided the stepsize is initialized properly" -- this should be stochastic?

**Limitations:**

Yes

---

> ### Author Rebuttal · Authors · 2023-08-07
>
> Thank you for your comments and constructive criticism.
>
> We address your concerns below:
> 1. On normalized gradient descent: We thank you so much for pointing out that the algorithm of Levy (2017) reduces to normalized gradient descent. We were not aware of this work and will include it. We'd like to point out that because we do the offline case rather than online learning, our analysis is much simpler, as it is not a reduction from AdaGrad. The main contribution of our work is not the NGD analysis but the DoWG algorithm, and NGD's analysis is used as motivation. We will give the correct citation for the result on NGD's convergence in the revision.
> 2. On weak vs strong adaptivity: We agree that the definition as-is is not very rigorous. We shall modify the definition per your suggestion, that an algorithm must also achieve the same rate as some baseline (e.g. Gradient Descent) that knows the convergence parameters. Moreover, we will add that we allow for polylogarithmic factors: this is not an artifact of our analysis, but is tight, as in general without knowledge of problem parameters we have to suffer at least a $\sqrt{\log \log \frac{D}{d_0}}$ or a $\sqrt{\log \frac{D}{d_0}}$ factor, depending on the class of methods, see [1, Theorems 4 and 6].
> 3. On "adapting to geometry": We agree that this term is too vague, despite its common usage in the literature on adaptive methods. Instead, we shall change it to adapting to smoothness/non-smoothness and/or regularity. We thank you for pointing this out.
> 4. On strong convexity: Indeed, due to the fact that NGD forces the effective stepsize to be 1/L, NGD does *not* adapt to strong convexity out of the box if we use a constant stepsize. Therefore a linear rate is not possible. We note that if problem constants are known, it is possible to extend to use an exponentially decreasing stepsize to get linear convergence out of NGD (see [2]). However, in general, it is easy to adapt to strong convexity (for any parameter-free algorithm with rates under smoothness and convexity) without knowledge of problem constants at all using the restarting scheme of [3]. The application of the restarting scheme of [3] to NGD/DoWG is straightforward, and we will include it in the appendix.
> 5. On the difficulty of obtaining the same results in the online learning setting: We agree that this would be instructive to include, the main difference is that in the offline setting, the adversary cannot change the function arbitrarily while just preserving the norm bound, as in the lower bound of [4]. Instead, the iterates have to come from the same function in response to the sequence of iterates picked by the algorithm. This allows for much improved rates. We will mention this in detail.
>
> [1] Konstantin Mishchenko and Aaron Defazio. "Prodigy: An Expeditiously Adaptive Parameter-Free Learner"
> [2] Damek Davis, Dmitriy Drusvyatskiy, Kellie J. MacPhee, Courtney Paquette. "Subgradient methods for sharp weakly convex functions".
> [3] Renegar, James, and Benjamin Grimmer. "A Simple Nearly Optimal Restart Scheme For Speeding Up First-Order Methods" Foundations of Computational Mathematics, vol. 22, no. 1, Mar. 2021, pp. 211–56. Crossref, https://doi.org/10.1007/s10208-021-09502-2.
> [4] Ashok Cutkosky, Kwabena Boahen. "Online Convex Optimization with Unconstrained Domains and Losses"

---

> > ### Comment · Reviewer_oAov · 2023-08-15
> > **Thanks**
> >
> > Thanks for the detailed response. My main concerns have been addressed.
> >
> > > The application of the restarting scheme of [3] to NGD/DoWG is straightforward, and we will include it in the appendix.
> >
> > The authors are welcome to include it, but I don't think the restart scheme extension is necessary. Mentioning that "universality" often includes the strongly-convex case, but does not apply to normalized GD (and would require an extension) should be  sufficient.
> >
> > ---
> >
> > (Minor point; please prioritize other responses)
> > Could you clarify the following?
> >
> > > On strong convexity: Indeed, due to the fact that NGD forces the effective stepsize to be 1/L, NGD does not adapt to strong convexity out of the box if we use a constant stepsize. Therefore a linear rate is not possible.
> >
> > If the effective step-size $\eta/\Vert\nabla f\Vert$ was forced to be 1/L, wouldn't the algorithm reduce to GD with step-size 1/L and get linear convergence?

---

> > > ### Author Response · Authors · 2023-08-16
> > >
> > > Thank you for following up with us. We are happy that your main concerns have been addressed.
> > >
> > > 1. (On universality) We will add that universality includes the strongly-convex case but doesn't apply to NGD. We agree it is important to point that out.
> > >
> > > 2. (On NGD) Because the effective stepsize stabilizes, this means that we cannot get much better than $\| \nabla f \| \approxeq \eta L$. What we observe (Figure 2 in the paper) is that when the gradient norms just keep oscillating. That is, the algorithm does a step that decreases the gradient norm, which results at the next step in a much larger effective stepsize, this in turn is too large (larger than the threshold $2/L$) and causes divergence, which forces the effective stepsize at the next iteration to be smaller, and so on. In this way, the effective stepsize oscillates around $2/L$, while at the same time not enabling linear convergence.

---

### Official Review · Reviewer_qfrF · 2023-07-11

**Soundness:** 4 excellent
**Presentation:** 4 excellent
**Contribution:** 3 good
**Rating:** 7
**Confidence:** 3

**Summary:**

This paper considers the problem of optimizing a convex function over a convex, closed, and possibly compact set $\mathcal{X}$. In particular, they are interested in finding a first-order method which is (i) universal (i.e., the same algorithm can be used when the objective is Lipschitz or smooth), (ii) parameter-free (i.e., converges without any hyperparameter tuning w.r.t. problem parameters) (iii) has no search subroutines, and (iv) strongly-adaptive (i.e., preserves the convergence rate of an optimally-tuned GD without tuning). The authors propose an algorithm, Distance-over-Weighted-Gradients, which (essentially) achieves all of these desired properties. The two caveats are (i) the convergence rates have an extra log factor not present in optimally-tuned GD, and (ii) the convergence rates depend on the diameter of the constraint set $\mathcal{X}$ instead of on $D_0 = || x_0 - x^* ||$. Towards establishing this result, the authors prove that normalized GD universal but not parameter-free.


**Strengths:**

This paper provides an algorithm with the remarkable properties of being universal, parameter-free, (nearly) strongly-adaptive, and not requiring a search subroutine. Unless I have misunderstood, this seems to be the first such algorithm satisfying all of these properties simultaneously. Moreover, the analysis is well-written and easy to follow. I also found the discussion of the universal properties of normalized GD to be quite instructive and interesting. Overall, I think this is a very nice paper.


**Weaknesses:**

Much of the analysis appears to rely on or extend arguments from Ivgi et al., (2023). Further, the results in this paper are restricted to the deterministic setting, while the results of Ivgi et al. hold also in the noisy setting (albeit, assuming uniformly bounded stochastic gradients, and thus the results are restricted to the Lipschitz setting). Despite this, I still think the results in this paper are quite nice.


**Questions:**

The results of Ivgi et al. (2023) hold also in the stochastic setting (assuming stochastic gradients are uniformly-bounded). What are the main barriers to extending your results to the stochastic setting? Is there any similar result that you can obtain when, e.g., the noise of stochastic gradients is uniformly bounded? It might be beneficial to the paper to add a discussion on why extending to this setting is difficult.

In the introduction of your paper, you mention that the constraint set $\mathcal{X}$ is “(potentially) compact“, However, your main results (Theorems 5 and 6) assume that the diameter $D$ of $\mathcal{X}$ is finite (but unknown). Is it possible to extend these results to the unconstrainted case where $\mathcal{X} = \mathbb{R}^d$?

---

> ### Author Rebuttal · Authors · 2023-08-07
>
> Thank you so much for your positive evaluation of our work.
>
> 1. On the stochastic setting: We agree that the stochastic case is important and merits exploration on its own, but we note that the results on the convergence of DoG in the stochastic case are not parameter-free, as they require knowledge of the Lipschitz constant on an unknown set. We have tried doing similar (non parameter-free) theory as in DoG, but unfortunately there is some added difficulty over the analysis of DoG. The main difference is that in DoG, the bound on the Lipschitz constant $G$ is used to apply concentration and obtain that the cumulative noise is small. Because in DoWG we may have no such bound, we cannot apply the same concentration inequality, at least in the smooth setting. We will add a more thorough discussion on this in the paper.
> 2. Unbounded domains: It is possible to extend DoWG to work for unbounded domains by using the same "taming" trick as in DoG: that is, dividing the stepsize by a logarithmic factor. We have carried out the analysis and everything works. The reason we did not include this algorithm is that it does not perform well in practice. We will include it in the appendix should the paper be accepted.

---

> > ### Comment · Reviewer_qfrF · 2023-08-18
> > **Thanks**
> >
> > Thank you for your response. I've read the discussion with the other reviewers, and will maintain my original score. In the next revision of the paper, I would recommend that the authors include their results on rates which depend only on $D_0$, as I agree with Reviewer WSp1 that these results are interesting (even if practical performance is not as good as the other step-size).

---

### Official Review · Reviewer_Uwk6 · 2023-07-11

**Soundness:** 3 good
**Presentation:** 3 good
**Contribution:** 2 fair
**Rating:** 5
**Confidence:** 3

**Summary:**

This paper introduces a new algorithm for optimization problems. The paper introduces DoWG (Distance over Weighted Gradient) which is a simple extension of a previous work DOG (Distance over gradient). It is a parameter-free gradient optimization method where the step size is automatically adjusted to the function being optimized over the course of the optimization.

This works also provides justification of why DoWG produces close to the optimal convergence that can be achieved by Normalized Gradient Descent (NGD) for either Lipschitz or smooth functions. As part of this claim the authors also show a new derivation of the optimal convergence for NGD on smooth functions.

Finally the authors show results on training Deep Nets for vision and these demonstrate the while DoWG performs better than other parameter-free methods it doesn't attain the level of accuracy achieved by momentum-based methods such as ADAM.

**Strengths:**

## Defining Universality

The authors have very systematically defined their concept of a Universal Gradient Descent-based optimization algorithm. In fact as part of this definition of Universality they have computed the optimal convergence rate of NGD for smooth functions which appears to be a new result although it is not a very surprising result.

## Discussions of NGD

The paper provides some very strong intuition about how NGD is self-stabilizing. The discussion and example in the paper provides good insight about how parameter-free methods work in general.


**Weaknesses:**

## Comparison to DOG

The proposed algorithm itself is only slightly different than the existing DOG algorithm and the intuition of why it was proposed is not that clear. The paper mentions that DoWG gives higher weight to later gradients, but why is this important? They do also demonstrate on line 263 that DoWG step sizes could be larger than DOG's and this seems reasonable, but overall the motivation for the improvement is not clear.

Now the paper does show that DoWG has optimal NGD convergence for Lipschitz functions, but it points out that the same holds for DOG. There is no discussion of whether optimal NGD convergence that is proven for smooth functions with DoWG also holds for DOG or not. If the authors could prove that DOG *doesn't* have optimal NGD convergence for smooth functions then their claim that DoWG is the first parameter-free Universal gradient descent algorithm without a search subroutine would be overall stronger.

## Performance Relative to ADAM

The authors should certainly be lauded for including a superior baseline result. However, the fact that ADAM does so much better than DoWG makes their work more of a theoretical curiosity.

It appears that the authors hurried through the final evaluation and included very limited results. The previous work that introduced DOG had results showing both fine-tuning as well as training a model from scratch. I would encourage the authors of the current work to do something similar as well as introduce variants such as L-DOWG.



**Questions:**

- Does DOG have optimal NGD convergence for smooth functions? Have the authors found a reason why this is not the case or why this would be unlikely and could provide a justification?

- Have the authors considered L-DOWG (layer-wise DOWG similar to L-DOG)?

- The DOG paper (https://arxiv.org/pdf/2302.12022.pdf) in Table 2 page 14 show results for DOG that are superior than ADAM for training a model from scratch. Why could the authors not reproduce those results as a baseline?

**Limitations:**

Not applicable.

---

> ### Author Rebuttal · Authors · 2023-08-07
>
> We thank you very much for your review.
>
> Please find our rebuttal below:
> 1. On motivation for DoWG: "The paper mentions that DoWG gives higher weight to later gradients, but why is this important?" The practice of assigning higher weights to later gradients is crucial to adaptivity to the loss curvature. For instance, if we start in a highly curved region where the gradients are large, a method that doesn't put a higher weight on the recent gradients would be stuck with small stepsizes for the rest of the training, even if it enters a flat region. This seems to be the reason why Adam works so well: it forgets older gradients at an exponential rate, as the gradient norm estimator in Adam can be written as $\hat{v}_T = \sum_{t=0}^{T} \beta_1^t g_{T-t}$. The same holds for the recently proposed Lion optimizer. The motivation is that as the optimization process continues, we get closer and closer to a minimizer and thus recent gradients give a more accurate direction to the minimizer.
> 2. On whether DoG can also converge for smooth objectives: at the time of writing the paper, DoG had no convergence guarantee in the smooth setting. In the most recent version of the DoG paper, the authors show that a certain iterate averaging strategy can yield smooth rates for DoG. However, the rate they show is $\mathcal{O} \left ( \frac{L D^2}{T} \left [ \log_+ \frac{D}{r_{\epsilon}} \right ]^2 \right )$ compared to DoWG's $\mathcal{O} \left ( \frac{L D^2}{T} \log_+ \frac{D}{r_{\epsilon}} \right )$, thus DoWG obtains a superior guarantee without changing the averaging strategy. This might be because DoWG takes larger, more aggressive stepsizes than DoG. Thus, while DoWG may not be the only algorithm that achieves universality, it is the best such algorithm in the literature.
> 3. We have considered L-DoWG, and an additional coordinate-only version of DoWG. We have made these implementations available online, but cannot link them due to anonymity. However, we were already short on space for the submission and thus didn't include them. Should the submission be accepted, we shall use the extra page to add in both algorithms and their motivation.
> 4. On superiority to Adam: Table 2 in the DoG paper shows that DoG with a polynomial decay averaging strategy performs better than Adam, but vanilla DoG does not. We reproduced a similar experiment in our paper, without the polynomial decay averaging. The results in our paper match those of Table 2 without polynomial decay averaging.

---

> > ### Comment · Reviewer_Uwk6 · 2023-08-20
> > **no major change based on response**
> >
> > Thanks for the response.
> > - For 4. It would help to list exactly where in the main text or supplementary material you have results showing better accuracy than Adam for Imagenet or CIFAR. (As shown in the DoG paper with polynomial decay averaging).

---

> > > ### Author Response · Authors · 2023-08-20
> > >
> > > The experiments in our paper do not show superiority of either DoG/DoWG over Adam because we do not use polynomial decay averaging. What we meant to say is that we reproduce that vanilla DoG (without polynomial decay averaging) is worse than Adam.

---

> > > > ### Comment · Reviewer_Uwk6 · 2023-08-20
> > > >
> > > > Could the authors be very precise. Did they ever run an experiment with polynomial decay averaging as was done in the DoG paper in their current work. How did the results compare to Adam? If they didn't run this experiment, why not?

---

> > > > > ### Author Response · Authors · 2023-08-20
> > > > >
> > > > > No, we did not run any experiments that included polynomial decay averaging. We used the vanilla DoWG/DoG for comparison against Adam. Using polynomial decay averaging requires passing a $\gamma$ parameter for the averaging, and we wanted to evaluate DoWG/DoG as parameter-free methods against Adam. We can do experiments that include polynomial decay averaging if they would be more instructive.

---

> > > > > > ### Comment · Reviewer_Uwk6 · 2023-08-20
> > > > > >
> > > > > > Repeating the prior experiment with polynomial decay averaging would be instructive to the extent that it would assure us that there is at least one scenario in which the new work does better than Adam.
> > > > > >
> > > > > > Also, if a paper builds very closely on top of a prior work it is expected (but not required) to repeat all the experiments in that prior work.

---

### Official Review · Reviewer_WSp1 · 2023-07-20

**Soundness:** 3 good
**Presentation:** 2 fair
**Contribution:** 2 fair
**Rating:** 6
**Confidence:** 4

**Summary:**

The paper proposes a new algorithm, DoWG, which modifies DoG (Ivgi et al., 2023) by utilizing the weighted sum instead of the usual sum for the squared gradients.
Convergence is established for convex and convex & smooth deterministic settings, under the assumption of a (possibly unknown) bounded domain.
The method is adaptive to the Lipschitz/smoothness constant and is parameter-free in the sense that the bound of the domain may be unknown.
Experiments comparing DoWG to several adaptive methods are presented.
Additionally, a complementary result for convex & smooth NGD establish that NGD is weakly adaptive to the diameter of the problem.

**Strengths:**

1. The modification from DoG to DoWG is appealing as it results in larger stepsizes, potentially leading to faster convergence.
2. A (limited) set of experiments is provided, demonstrating positive empirical evidence for the effectiveness of the DoWG method.

**Weaknesses:**

3. It is important to highlight that the convergence of DoWG is established exclusively in the deterministic setting, making it weaker than DoG's convergence, which extends to both deterministic and stochastic settings (note that AdaGrad-Norm results also apply to the stochastic setting). The paper should explicitly emphasize this difference and consider exploring DoWG's performance in the stochastic setting.
4. The theoretical results for DoWG hold significance when dealing with unknown domain bounds. When the domain bound is known, AdaGrad-Norm with stepsizes $\eta_t=D/\sqrt{\sum_t \lVert \nabla f(w_t) \rVert^2}$ has tighter bounds for both convex and convex & smooth settings.
5. The term "parameter-free method" typically used for rates that depends on the distance to the comparator or the minimizer. In this context the convergence of DoWG is not truly parameter-free as it is relevant only for bounded domains and does not improve with a good initialization.

**Questions:**

6. Can DoWG achieve similar convergence results to DoG in the stochastic setting? If it is yet established, what is the additional difficulty with respect to DoG?
7. In the deterministic setting, would DoG achieve similar convergence results to DoGW, or was the modification to DoWG necessary to address specific issues in that setting?
8. What is the motivation for the problem of deterministic optimization with unknown bounded domain? Corollary 1 of Ivgi et al. (2023) mentions two-stage stochastic programming as an application, but is the deterministic counterpart of interest as well?

Overall, while DoWG shows promise and may yield novel convergence properties, I find the current established guarantees to be partly insufficient compared to previous work on adaptive and parameter-free methods.

Edit: Per the authors response regarding the unbounded case with adaptivity to $D_0$ and the improved log factor with respect to the smooth result of DoG (unknown at the time of submission), i raise my score.

---

> ### Author Rebuttal · Authors · 2023-08-07
>
> Thank you very much for your positive evaluation and constructive criticism of our work. We agree that DoWG's main strength is that it allows for much larger stepsizes than DoG both in theory and practice.
>
> We now address the weaknesses and questions:
> 1. On the stochastic case: We agree that the stochastic case is important and merits exploration on its own, but we note that the results on the convergence of DoG in the stochastic case are *not* parameter-free, as they require knowledge of the Lipschitz constant $G$ on an unknown set. We have tried doing similar (non parameter-free) theory as in DoG, but unfortunately there is some added difficulty over the analysis of DoG. The main difference is that in DoG, the bound on $G$ is used to apply concentration and obtain that the cumulative noise is small. Because in DoWG we may have no such bound, we cannot apply the same concentration inequality, at least in the smooth setting.
> 2. Comparison with AdaGrad-Norm: We note that if the domain bound is known, we can simply plug it into the initialization of DoWG, and DoWG will not change it: This is a consequence of Theorems 5 and 6, where if we put $r_{\epsilon} = D$ (i.e. set the seed initialization to $D$) then $\log_+ \frac{D}{r_{\epsilon}} = \log \frac{D}{r_{\epsilon}} + 1 = \log 1 + 1 = 1$. Therefore, the rate becomes exactly the same as AdaGrad-Norm, with no additional logarithmic factor.
> 3. Convergence of DoWG for unbounded domains: We note that if we slightly reduce the DoWG stepsize, by using $\eta = \frac{r_t^2}{v_t} \frac{1}{\log_+ \frac{v_t}{v_0}}$, this allows for convergence rates that depend only on $D_0$ rather than the domain bound $D$. The main reason we did not include this variant is that it did not perform well in practice. We shall include this variant and its theory in the final paper, should it be accepted.
> 4. Comparison with DoG in the smooth setting: In the most recent version of the DoG paper, the authors show that a certain iterate averaging strategy can yield smooth rates for DoG. However, the rate they show is $\mathcal{O} \left ( \frac{L D^2}{T} \left [ \log_+ \frac{D}{r_{\epsilon}} \right ]^2 \right )$ compared to DoWG's $\mathcal{O} \left ( \frac{L D^2}{T} \log_+ \frac{D}{r_{\epsilon}} \right )$, thus DoWG obtains a superior guarantee without changing the averaging strategy. This might be because DoWG takes larger, more aggressive stepsizes than DoG.
> 5. Motivation for unknown domain:  Our main motivation in DoWG is to apply this theory in practice. We note that in practice, a combination of reasonable initialization and bounded updates (enforced through e.g. gradient clipping) lead to a domain bound on the iterate that is unknown. For example, [1] show that neural networks with Gaussian initialization and bounded data norms are semi-smooth and generate a sequence of bounded iterates, with the domain bound being a quite complicated function of the neural network's architecture. While the objective of [1] is non-convex, our theory can instead be applied to convex neural networks with similar properties, e.g. Gated Linear Networks common in data compression algorithms [2].
>
> [1] Zeyuan Allen-Zhu, Yuanzhi Li, Zhao Song. A Convergence Theory for Deep Learning via Over-Parameterization. arXiv:1811.03962.
> [2] Joel Veness, Tor Lattimore, David Budden, Avishkar Bhoopchand, Christopher Mattern, Agnieszka Grabska-Barwinska, Eren Sezener, Jianan Wang, Peter Toth, Simon Schmitt, Marcus Hutter. Gated Linear Networks. arXiv:1910.01526

---

> > ### Comment · Reviewer_WSp1 · 2023-08-13
> >
> > I thank the authors for their elaborate comment.
> >
> > 1. I strongly suggest to include the result that depends only on $D_0$, including a discussion of the result in the main body. Calling the method "parameter-free" without adaptivity to the distance form the minimizer (comparator in online learning) might be misleading.
> > 2. The same goes for support for unbounded domains. It is fine to have a theoretical method and an "in-practice" small modification of the method. Those two guarantees makes the theoretical setting much less restricted.
> > 3. Did the authors considered a version that use the Lipschitz parameter for the stochastic setting? Will this modification solve the stochastic setting? A common distinction is that a method is parameter-free if it is adaptive to the distance from the minimizer (or comparator, i.e., in parameter-space), and scale-free if no knowledge of the Lipschitz constant is needed (e.g. [1,2]).
> >
> > [1] Orabona, Francesco, and Dávid Pál. "Open problem: Parameter-free and scale-free online algorithms." Conference on Learning Theory. PMLR, 2016.
> > [2] Orabona, Francesco, and Dávid Pál. "Scale-free online learning." Theoretical Computer Science 716 (2018): 50-69.

---

> > > ### Author Response · Authors · 2023-08-14
> > >
> > > Thank you for your quick response.
> > >
> > > 1. (On proof with $D_0$ and unbounded domains). We shall include the result on the variant of DoWG with dependence on the initial distance $D_0$ in the final paper. The result is not very difficult to derive, and we include the main lemma here for completeness. The main idea is that by dividing the stepsize by a running logarithmic factor, we can prove the iterates stay bounded by a constant multiplied by the initial distance $d_0$. Then we can just apply the ordinary DoWG proof (at the cost of an additional logarithmic factor only). The proof follows follows [1].
> > >
> > > Lemma 1 (Stability). Suppose that $r_0 \leq d_0$. Then the iterates of DoWG with stepsizes $\eta_t = \frac{\bar{r}_t^2}{2 \sqrt{v_t}} \frac{1}{\log \frac{2v_t}{v_0}}$ satisfy $\bar{d}_t \leq 16 d_0$ and $\bar{r}_t \leq 16 d_0$ for all $t$.
> > >
> > > Proof. We have by convexity
> > >
> > > $$ d_{k+1}^2 - d_k^2 \leq \eta_k^2 |{g_k}|^2 $$
> > >
> > > Summing up from $k=1$ to $k=t$ we get by [1, Lemma 6]
> > >
> > > $$ d_t^2 - d_1^2 \leq \sum_{k=1}^{t} \eta_k^2 |{g_k}|^2 = \sum_{k=1}^{t} \frac{ {\bar{r}\_k}^4 }{v_{k-1}} \frac{|{g_k}|^2}{4 \log^2 \left( \frac{2v_{k-1}}{v_0} \right)} \leq \frac{\bar{r}\_t^2}{4} \sum{k=1}^{t-1} \frac{v_k - v_{k-1}}{v_{k-1} \log_+^2 \left( \frac{v_{k-1}}{v_0} \right)} \leq \frac{1}{4} \bar{r}\_t^2 $$
> > >
> > > Thus we have $d_{t+1}^2 \le d_1^2 + \frac{\bar{r}t^2}{4}$. Now suppose in the way of induction that $\bar{r}\_t^2 \leq 8 d_1^2$, then applying the last equation we get $d\_{t+1}^2 \leq d_1^2 + \frac{8}{4} d_1^2 = 3 d_1^2$. Taking square roots gives $d{t+1} \leq \sqrt{3} d_1$. Subsequently, by the triangle inequality we have
> > >
> > > $$ |{x_{t+1} - x_1}| \leq |{x_{t+1} - x_{}}| + |{x_{} - x_1}| \leq \left[ \sqrt{3} + 1 \right] d_1 $$
> > >
> > > Squaring both sides gives $|{x_{t+1} - x_1}|^2 \leq \left( \sqrt{3}+1 \right)^2 d_1^2 \leq 8 d_1^2$. It follows that $\bar{r}\_{t+1}^2 \leq 8 d_1^2$, and the induction thus gives us $\bar{r}_t^2 \leq 8 d_1^2$ for all $t$. Next, observe that
> > >
> > > $$ d_1 = |{x_1 - x_{}}| \leq |{x_0 - x_1}| + |{x_0 - x_{}}| = r_0 + d_0 \leq 2 d_0 $$
> > >
> > > It follows that $\bar{r}\_t \leq 16 d_0$ for all $t$. Finally, observe that the same analysis implies $d_t \leq 16 d_0$ for all $t$. Therefore, we get that the iterates stay in a bounded domain, and with a small modification of the main proof, we get the same result as the original paper, with $D$ replaced by $16 D_0$ and an additional log factor. We shall include the result in full in the final paper.
> > >
> > > 2. If we do have knowledge of the Lipschitz parameter (or the Lipschitz smoothness parameter in the smooth case) then we can do the stochastic case. We did not consider this for long because the algorithm is then no-longer parameter free. In the online learning setting, it is not possible to do away with knowledge of the Lipschitz parameter in the worse case, as the lower bound of Cutkosky and Boahen (2016) shows. This may not be the case for offline stochastic convex optimization.
> > >
> > > References:
> > > [1] Ivgi et al., DoG is SGD's best friend, 2023.

---

> > > > ### Comment · Reviewer_WSp1 · 2023-08-19
> > > >
> > > > I thank the authors for their response.
> > > >
> > > > Per the authors response regarding the unbounded case with adaptivity to $D_0$ and the improved log factor with respect to the smooth result of DoG (unknown at the time of submission), i raise my score.

---

### Official Review · Reviewer_jvtQ · 2023-07-23

**Soundness:** 4 excellent
**Presentation:** 4 excellent
**Contribution:** 4 excellent
**Rating:** 8
**Confidence:** 2

**Summary:**

The paper proposes a new optimization algorithm, DOWG, that does not rely on additional hyperparameter tuning or a line search subroutine. Interestingly, the paper also contains a proof and analysis regarding the behavior of NGD and shows that (1) NGD adapts to the smoothness of continuous loss surfaces and (2) NGD operates at the edge of the maximum stable learning rate for continuous loss functions. Finally, the paper presents two empirical studies on CIFAR10 with two different neural networks.

**Strengths:**

I am not an expert in optimization, but I thoroughly enjoyed reading this paper. The writing is clearly organized and accessible to readers with only some rudimentary knowledge of optimization.

Regarding related work, the paper carefully assigns credit to existing work and precisely describes how the current work stands out at each step along the way, which demonstrates a high level of expertise in the field.

The theoretical contribution seems solid, and the new algorithm seems to be well motivated and work well, when compared with other parameter-free algorithms. I strongly recommend this paper be accepted.

**Weaknesses:**

DOWG does not work as well as Adam with cosine annealing on both neural network problems. However, this is not very surprising because cosine annealing is extremely strong. I do not consider this to be a fatal issue.

**Questions:**

Nil

**Limitations:**

Nil

---

> ### Author Rebuttal · Authors · 2023-08-07
>
> Thank you so much for your very positive evaluation of our work. We agree that Adam with cosine annealing is an incredibly strong baseline, and we hope that future work can reveal a principled way of improving over that.

---

> > ### Comment · Reviewer_jvtQ · 2023-08-16
> >
> > Thanks. I've read the response and have nothing further to add.

---

### Decision · Program_Chairs · 2023-09-21

**Decision:**

Accept (poster)

**Comment:**

After the author responses and discussion phase, all reviewers seem happy with the paper. I would strongly recommend that the authors incorporate the issues brought up in the reviews and discussion phase into the final version.